# Small-scale fisheries catch more threatened elasmobranchs inside partially protected areas than in unprotected areas

Manfredi Di Lorenzo [1,6] ✉, Antonio Calò [2,6], Antonio Di Franco[1,6] ✉, Giacomo Milisenda[1], Giorgio Aglieri [1,2,3], Carlo Cattano [1,2,3], Marco Milazzo [2,3] & Paolo Guidetti[4,5]

Elasmobranchs are heavily impacted by fishing. Catch statistics are grossly underestimated due to missing data from various fishery sectors such as small-scale fisheries. Marine Protected Areas are proposed as a tool to protect elasmobranchs and counter their ongoing depletion. We assess elasmobranchs caught in 1,256 fishing operations with fixed nets carried out in partially protected areas within Marine Protected Areas and unprotected areas beyond Marine Protected Areas borders at 11 locations in 6 Mediterranean countries. Twenty-four elasmobranch species were recorded, more than one-third belonging to the IUCN threatened categories (Vulnerable, Endangered, or Critically Endangered). Catches per unit of effort of threatened and data deficient species were higher (with more immature individuals being caught) in partially protected areas than in unprotected areas. Our study suggests that despite partially protected areas having the potential to deliver ecological benefits for threatened elasmobranchs, poor small-scale fisheries management inside Marine Protected Areas could hinder them from achieving this important conservation objective.

Elasmobranchs (sharks, skates, and rays) are among the megafauna that are most threatened by fishing[1–4]. Often caught as bycatch, many elasmobranchs have increasingly become target species over the last few decades due to the increased demand for elasmobranch products (especially fins and meat)[5]. Global catch of elasmobranchs continued to increase, reaching a peak of 900,000 tonnes per year in 2003[6], and have subsequently declined as a result of overfishing[7]. Global statistics of catches are presumably underestimated by a factor of three or four, as they do not consider Illegal, Unreported, and Unregulated (IUU) catches[8]. The substantial exploitation of elasmobranchs has had serious consequences, and recent estimates suggest a 71% decline in the global abundance of oceanic sharks and rays since 1970[4]. This steep decline in elasmobranch abundance is exacerbated by their peculiar life-history traits (i.e., large size, slow growth rate, late maturity, and low fecundity) which makes them particularly vulnerable to fishing disturbance. As a result, about one-third of elasmobranch species are classified as threatened with extinction (Vulnerable, Endangered, or Critically Endangered) in the Red List of the International Union for the Conservation of Nature (IUCN)[4,9].

Small-scale fisheries (SSF) employ 90% of the world's fishers and contribute to over 50% of global marine species catch worldwide[10]. Yet,

[1]Stazione Zoologica Anton Dohrn, Department of Integrative Marine Ecology, Sicily Marine Center, Lungomare Cristoforo Colombo (complesso Roosevelt), 90149 Palermo, Italy. [2]Department of Earth and Marine sciences (DiSTeM), University of Palermo, Via Archirafi 20-22, 90123 Palermo, Italy. [3]CoNISMa, Piazzale Flaminio 9, 00196 Rome, Italy. [4]Department of Integrative Marine Ecology (EMI), Stazione Zoologica Anton Dohrn–National Institute of Marine Biology, Ecology and Biotechnology, Genoa Marine Centre, 16126 Genoa, Italy. [5]National Research Council, Institute for the Study of Anthropic Impact and sustainability in the Marine Environment (CNR-IAS), Via de Marini 6, 16149 Genova, Italy. [9]These authors contributed equally: Manfredi Di Lorenzo, Antonio Calò, Antonio Di Franco. ✉ e-mail: manfredi.dilorenzo@libero.it; antonio.difranco@szn.it

SSF have been systematically overshadowed by the large-scale fisheries sector and largely understudied[11]. Most of the available information on the status of elasmobranch species and fishing impact is obtained from the large-scale fishery sector[12–15]. Much less is known about elasmobranchs and their interactions with SSF in coastal areas, which represent data-poor systems[5,10]. SSF are generally acknowledged as being less impactful and more sustainable than large scale fisheries[16], however, some studies have highlighted that these fisheries can represent a significant threat for a number of vulnerable marine vertebrates (e.g., marine mammals and sea turtles[17–19]) that are caught as bycatch. To date, the potential impact of SSF on elasmobranch species has been poorly understood[11] and existing studies are scarce and geographically limited[20–23].

Growing concerns over the global status of elasmobranchs and their ongoing depletion[3,9,24] have led to an increase in conservation efforts over the last decade to protect them and mitigate against their decline[25]. However, questions remain regarding what strategies to apply to best protect them. Marine protected areas (MPAs), including fully protected areas (FPAs) and partially protected areas (PPAs), are effective tools to protect marine biodiversity[26–28], whilst at the same time, have the potential to improve fishers' wellbeing[29–31], reconciling conservation and fisheries goals[32]. MPAs are generally considered effective in protecting species with limited movements[33,34], but recent evidence pointed out their potential to conserve mobile and long-lived predators, including elasmobranchs[25,35]. Several very large FPAs have been established worldwide and are being promoted as a tool for conservation and recovery of pelagic species (including elasmobranchs)[36,37]. The recent designation of large FPAs has greatly helped in achieving global protection targets[38]. On a global scale, around 29% of the total protected ocean area (corresponding to ~7 million km²) is dedicated to elasmobranch conservation. This is most likely considerably too small to achieve elasmobranch conservation goals, especially considering that only 2.8% of the world's oceans are fully protected[25]. Although criticisms exist regarding their effectiveness[38,39], potential positive effects of large FPAs on elasmobranch species have recently been highlighted[4,40,41]. Most existing FPAs, however, are small, and therefore potentially unable to deliver full conservation benefits for large or mobile elasmobranchs. Yet, positive effects for these species may arise through the protection of critical habitats for reproduction and feeding[42]. To date, these benefits for elasmobranchs remain relatively unknown[43]. As MPAs can trigger virtuous fishers' attitudes and recognition of the need to cooperate with MPA managers and scientists, they represent an ideal incubator for wider successful cooperation for the conservation of endangered species, including elasmobranchs (see e.g.,[44]). Most MPAs globally are multiple-use PPAs, where some regulated human activities are permitted[28,45]. PPAs generally cover most of the surface area of multiple-use MPAs, and under certain circumstances have been proven to deliver ecological benefits to coastal fish[46], however limited evidence exists regarding their effectiveness in protecting elasmobranch species.

The Mediterranean Sea is an important elasmobranch biodiversity hotspot[47]. However, its long history of human exploitation[48], including fishing pressure, along with habitat loss and degradation[49] have led to a steep decline in elasmobranchs[47], with the threat status of Mediterranean elasmobranchs worsening more rapidly than the global status[50]. In fact, a marked regional decline has been highlighted for large predator shark species[51] and smaller commercially important meso-predator species[21,52,53]. SSF in the Mediterranean Sea are multi-species fisheries that mainly target teleost fishes; elasmobranchs are not usually targeted, but when fished are generally retained and sold. Accounting for 83% of fishery vessels in the Mediterranean Sea[54], SSF have been found to impact elasmobranch populations (e.g.,[23,55]). Although the Mediterranean Sea hosts many coastal multiple-use MPAs, mostly PPAs[56], no studies to date have focused on the potential role of PPAs to protect the elasmobranch species that were once widespread throughout the Mediterranean[57].

Our study was designed to fill in these critical information gaps and to characterize elasmobranch assemblages caught by SSF within the PPAs and/or in the surrounding unprotected areas (UPAs) of 11 Mediterranean MPAs (Fig. 1; Supplementary Fig. 1). Due to the probable higher fishing pressure in UPAs than in PPAs given the fishing restrictions generally imposed in PPAs (fishing effort data of SSF are not generally available in the Mediterranean as SSF vessels

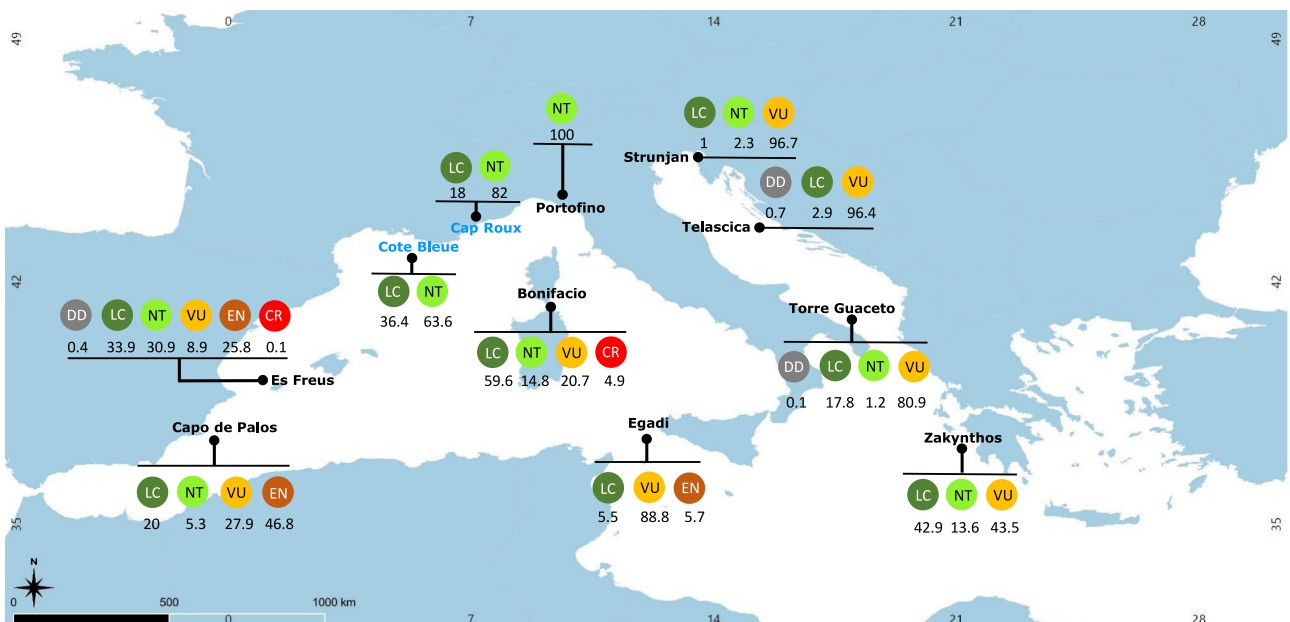

**Fig. 1 | Study locations.** Numbers report the percentage of elasmobranchs by IUCN categories caught (i.e., Biomass CPUE data) at each location. Names in black indicate locations where a multiple-use MPA is present and where catches were monitored both within the partially protected areas (PPAs) and in nearby unprotected areas (UPAs); names in blue indicate locations where the MPA is entirely covered by fully protected areas (FPAs) and where catches were monitored only in UPAs. CR Critically Endangered, EN Endangered, VU Vulnerable, NT Near Threatened, LC Least Concern, DD Data Deficient. Source data are provided as a Source Data file.

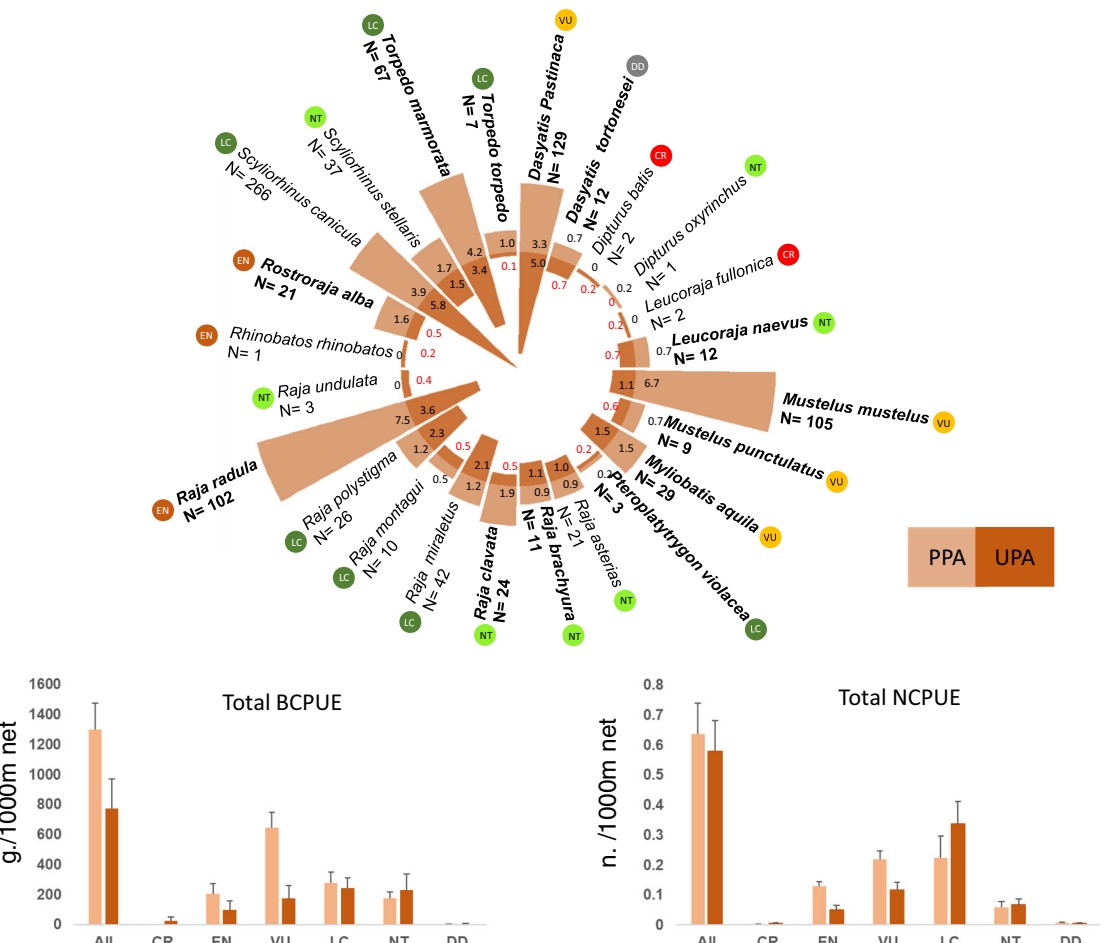

**Fig. 2 | Percentage of fishing operations in which elasmobranch species have been recorded in partially protected areas (PPAs) and in unprotected areas (UPAs).** 'N' indicates the overall number of individuals caught for each species. The species in bold were caught (as NCPUE) mostly in PPAs. The barplots show BCPUE (bottom-left) and NCPUE (bottom-right) (mean ± se) in PPAs and in UPAs. CR Critically Endangered, EN Endangered, VU Vulnerable, NT Near Threatened, LC Least Concern, DD Data Deficient. Source data are provided as a Source Data file.

are not equipped with tracking devices) we hypothesize that elasmobranch catch per unit of effort (CPUE) is higher in PPAs than UPAs. We use photo-sampling and subsequent image analysis of catches from fixed nets, the gear most used in Mediterranean SSF, to compile an extensive database covering 1256 fishing operations in six countries, along with the regional IUCN red-list assessments to: (1) assess the interaction between SSF and coastal elasmobranch species; (2) investigate potential differences in species biomass CPUE (BCPUE, grams per 1000 m of net), and abundance CPUE (NCPUE, number of individuals per 1000 m of net) between PPAs and UPAs in nine out of eleven locations (see material and methods section for more details), disentangling the effect of protection from a set of potential covariates (for example, chlorophyll a and sea surface temperature. See Supplementary Table 1). Our study provides evidence on the role of PPAs in protecting coastal elasmobranch species and highlights that SSF may represent a threat for these species, also inside Mediterranean MPAs, suggesting the critical need for careful management measures.

## Results and discussion
### Elasmobranch catches by SSF are mostly represented by Threatened species
Using landing data from 1256 fixed net operations (for a total of 737.71 km of nets deployed) carried out to depths of 150 m, we recorded 892 elasmobranch individuals belonging to 24 species (four demersal sharks and twenty batoid species) (Fig. 2; Supplementary

Tables 2, 3a, b), representing almost one third of the total elasmobranch species living in the Mediterranean Sea up to depths of 800 m[58]. Considering all the species caught during the SSF operations, elasmobranchs accounted for 2.4% of the overall NCPUE and 6.4% of the overall BCPUE. According to the last assessment for the Mediterranean IUCN red list, more than one third of the species caught are categorized as Threatened [THR] (including Critically Endangered, Endangered and Vulnerable categories) (Fig. 2; Supplementary Table 2). Blue Skate (*Dipturus batis* Linnaeus, 1758) and Shagreen Ray (*Leucoraja fullonica* Linnaeus 1758) were the only species caught that are listed as Critically Endangered [CR]. Three species are listed as Endangered [EN] (Rough ray, *Raja radula* Delaroche 1809; Common Guitarfish, *Rhinobatos rhinobatos* Linnaeus 1758; and White Skate, *Rostroraja alba* Lacepède 1803), four as Vulnerable [VU] (Common Stingray, *Dasyatis pastinaca* Linnaeus 1758; Common Eagle Ray, *Myliobatis Aquila* Linnaeus 1758; Common Smooth-hound, *Mustelus mustelus* Linnaeus 1758; and Blackspotted Smooth-hound, *Mustelus punctulatus* Risso 1827) and seven as Near Threatened [NT] (Long-nose Skate, *Dipturus oxyrinchus* Linnaeus 1758; Cuckoo Ray, *Leucoraja naevus* Müller & Henle, 1841; Mediterranean Starry Ray, *Raja asterias* Delaroche, 1809; Blonde Ray, *Raja brachyura* Lafont 1873; Thornback Ray, *Raja clavata* Linnaeus 1758; Undulate Ray, *Raja undulata* Lacépède, 1802; and Nursehound, *Scyliorhinus stellaris* Linnaeus 1758). The remaining species recorded are categorized as Least Concern [LC] and Data Deficient [DD] (Fig. 2, Supplementary Table 2). The greatest biomass per unit effort was for THR species

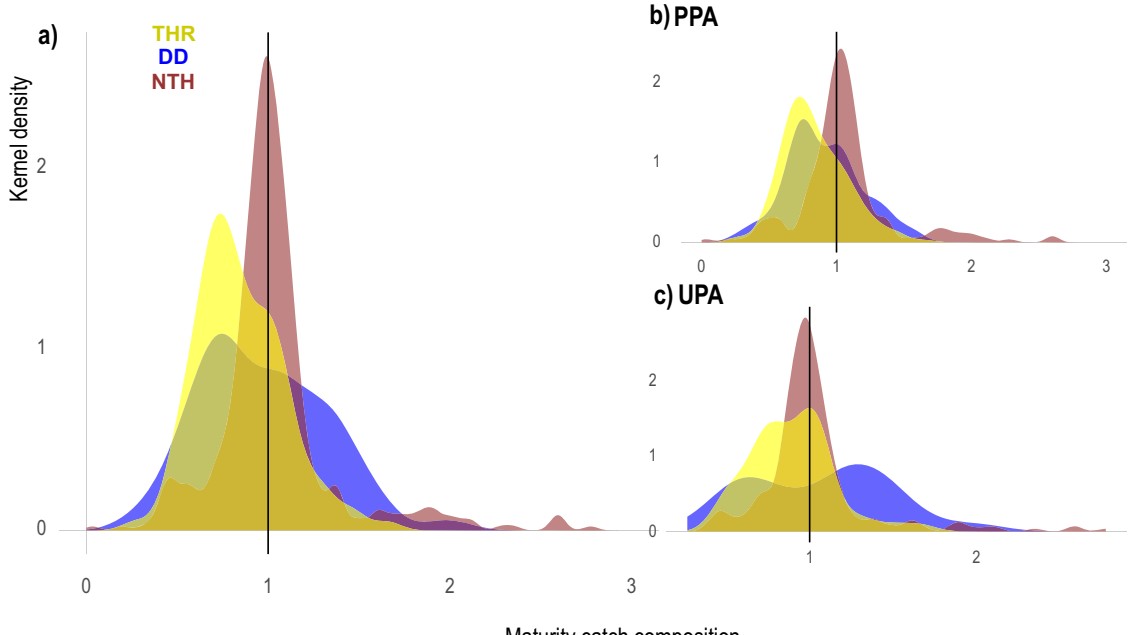

**Fig. 3 | Kernel distribution plots. a** Proportions of mature (standardized maturity index ≥1) and immature (standardized maturity index <1) specimens caught (data from PPAs and UPAs pooled) for the three extinction risk groups (TH = Threatened, NTH = Nonthreatened, DD = Data Deficient). **b** and **c** represent the break-down for PPAs and UPAs, respectively. The black vertical lines indicate maturity index corresponding to $L_{50}$.

(BCPUE 507.73 ± 74.3 g/1000 m, mean ± SE), and the least for DD species (2.5 ± 1.0 g/1000 m, mean ± SE).

Our study highlights that, as well as trawl and longline fisheries[2], SSF fixed nets also impact threatened elasmobranchs in the Mediterranean Sea[23,59]. This is very concerning as SSF account for most of the fishing vessels operating, not only in the Mediterranean Sea, but also worldwide[16,17,54,60]. Elasmobranch captures represented a relatively small percentage of the total catches in our study (both in terms of NCPUE and BCPUE); however, given the high number of elasmobranch species captured, out of the total living in the Mediterranean Sea, and the considerable proportion of THR species fished, a systematic assessment of SSF catches should be implemented. In fact, more than half of the species caught by SSF in five out of eleven study locations (Capo de Palos, Egadi Islands, Strunjan, Telascica and Torre Guaceto) are considered threatened with extinction. In line with past studies about temporal trends of demersal elasmobranchs presence in those areas[52,53], VU species were the most frequently caught. The low NCPUE and BPCUE recorded in our study could be due to multiple pressures affecting Mediterranean coastal ecosystems, with historical overfishing playing a major role, as has already been highlighted by previous studies[3] (see the case of *Mustelus* spp. decline in the Mediterranean Sea[52]). Generally, elasmobranch catches are difficult to assess due to low data availability, especially given the lack of species-specific reporting[50]. Elasmobranchs are often reported/sold using generic terms, such as "shark", "ray" or other vernacular names, making it hard to assess the decline of some species[7]. Indeed, a systematic assessment of SSF catches along Mediterranean coasts requires accurate taxonomic and morphometric information for landed specimens that also includes the discards of non-commercial species, including elasmobranchs, as this data is crucial to evaluate the actual impact of SSF on elasmobranchs[50] and to plan sound fisheries management. Even if species-specific reporting poses a cost in terms of capacities required (training people and/or technologies[61]), here we show that image analysis of SSF landings is an effective method to monitor catch because: (1) it is straightforward and can be performed by any (e.g. scientists, MPA staff, etc.)

using cheap technology; and (2) images represent a reliable record and allow retrospective accurate identifications by experts whenever necessary.

Kernel density distributions (probability peaks) suggested that immature individuals were caught more frequently than mature individuals for THR and DD species (Fig. 3, Supplementary Table 4), and a similar proportion of mature and immature individuals belonging to species in the non-threatened group (Near Threatened and Least Concern species: hereafter NTH) were caught (Fig. 3, Supplementary Table 4). This finding could be due to ray species' (the most represented group of NTH) sedentary behaviour throughout their life cycle[62], and the fact that they do not show habitat segregation between juveniles and adults. This pattern differs in sharks, like *Mustelus* spp. (belonging to THR), that, besides the reproductive season, when they show high site fidelity, generally have larger home ranges during the adult stage and move away from coastal areas where SSF operate[62].

Since most SSF operate in coastal areas encompassing important and often critical habitats for reproduction and juvenile survival (i.e., mating and nursery areas)[63–66], it is likely that recruitment overfishing (i.e., the rate of fishing above which the recruitment to the exploitable stock becomes significantly reduced) could be one of the causes leading slow-growing elasmobranch species to risk of extinction. Therefore, to effectively protect Mediterranean elasmobranchs, we suggest that: (i) in the absence of ad hoc assessments, DD species should be managed and monitored with the same conservation strategies adopted for the THR group, and (ii) like many teleost species[67], a minimum landing size should be set for Mediterranean elasmobranchs to ensure individuals survive to maturity, increasing the probability of reproduction before being caught.

**Higher elasmobranch catches per unit of effort inside partially protected areas**

Concerning the 9 locations used for the comparison between elasmobranch catches in PPAs and UPAs, 517 individuals were collected inside PPAs during 573 fishing operations (for a total of 385.51 km of nets deployed) and 358 individuals were collected in UPAs during 511

fishing operations (for a total of 352.20 km of nets deployed). The overall biomass of elasmobranchs recorded was higher for fishing operations in PPAs (487.53 kg) than in UPAs (223.07 kg). The mean BCPUE and NCPUE of elasmobranchs were higher inside the PPAs (1299.8 ± 177.6 g/1000 m., 0.63 ± 0.08 n/1000 m, mean ± SE, respectively; Fig. 2) than in UPAs (773.6 ± 197.3 g/1000 m., 0.57 ± 0.10 n/1000 m; Fig. 2). Generalized additive models for location, scale, and shape (GAMLSS) on BCPUE and NCPUE data suggested that 'protection' significantly affected both response variables, and although the probability of having elasmobranchs in SSF catches (i.e., the probability of fishing operations with at least one elasmobranch individual in the catch) was higher in UPAs than PPAs (BCPUE: $t_{value} = -2.95$, $p = 0.003$; NCPUE: $t_{value} = -2.82$, $p = 0.004$, "*logit*" part of the model, Supplementary Table 5, Supplementary Figs. 2b, 3b), the BCPUE and the NCPUE were found to be higher inside the PPAs ($t_{value} = -2.81$, $p = 0.001$; $t_{value} = -2.93$, $p = 0.003$ respectively; "*log*" part of the model, in Supplementary Table 5; Supplementary Figs. 2a, 3a). Among the environmental covariates considered, only 'chlorophyll a' showed a significant and positive relationship with both BCPUE and NCPUE (*log*: $t_{value} = 6.96$, $p = 0.001$, $t_{value} = 3.70$, $p = 0.001$ respectively; Supplementary Table 5), whilst BCPUE and sea surface temperature (SST) were inversely related (*log*: $t_{value} = 3.69$, $p = 0.001$; Supplementary Table 5). We also found a positive relationship between the overall human impacts index considered and NCPUE (*log*: $t_{value} = 2.62$, $p = 0.008$; Supplementary Table 5), while no relationship between the index and BCPUE was detected. These findings suggest that the positive relationship detected for NCPUE is mainly determined by the presence of juveniles and/or small-sized species (that contribute to the number of individuals, but less in terms of biomass) in areas with high human impacts. In the absence of additional factors being investigated, we hypothesize that the counterintuitive relationship between NCPUE and human impacts may be related to a combination of factors including the removal of top predators in highly impacted areas, potentially releasing meso-predators and/or juveniles[35,51]. This mechanism has been previously suggested[51], but the opposite has also been reported[68]. These studies however, referred to species exploited by large scale fisheries, and to the best of our knowledge no previous studies are available for SSF in coastal areas. GAMLSS explained 37.0% and 41.2% of the deviance for BCPUE and NCPUE, respectively (Supplementary Fig. 4). The positive effect of the protection on elasmobranch species found in this study has also been emphasized, using a multivariate perspective, by the pRDA performed on the species grouping into the 6 IUCN categories. In fact, the results of the pRDA highlighted that when considered singularly the Endangered and Vulnerable categories were highly correlated with the PPAs (Fig. 4; 999 permutations: $p < 0.001$ for both BCPUE and NCPUE).

Overall, higher numbers of immature individuals were caught both in PPAs and UPAs (Supplementary Fig. 5). The proportion of mature individuals did not differ between PPAs and UPAs for THR, NTH and DD groups (THR, $Z_{39} = 0.946$ $p = 0.998$; NTH, $Z_{91} = -1.242$ $p = 0.217$; DD, $Z_{32} = 0.432$ $p = 0.669$; Supplementary Fig. 5). On the contrary, the statistical analyses revealed that immature individuals of THR and DD species were more frequently captured inside PPAs (THR, $Z_{85} = -3.268$ $p = 0.0015$; DD, $Z_{29} = -2.006$ $p = 0.006$; Supplementary Fig. 5), whilst for the NTH group no statistical differences were found between protection levels ($Z_{86} = 0.748$ $p = 0.456$). The result for THR and DD species may indicate that PPAs could support the presence of juveniles. Many coastal areas have been altered and have deteriorated due to human impacts[69]; MPAs, instead, are zones where habitat protection/recovery is promoted, and where, potentially, healthy/recovered habitats can act as nurseries for elasmobranchs.

Given the higher BCPUE and NCPUE of elasmobranchs captured within the PPAs, our results suggest that PPAs may play an important role in protecting threatened elasmobranch species along the Mediterranean coast. This finding suggests that restrictions on human use

activities, as found in PPAs (reduced fishing effort, use of less-impacting fishing gears, and reduction of other sources of human disturbance in general, Supplementary Table 6), could help increase the density and biomass of these threatened species[15]. In addition to direct protection effects, indirect factors could also play a role. For instance, an increase in biomass of elasmobranch prey within the PPAs may attract elasmobranch species to these areas. The elasmobranch species we observed in SSF catches feed on a large spectrum of smaller marine organisms, such as crustaceans, molluscs, and fish[70,71], which all benefit from the 'reserve effect'[26]. Unfortunately, we lack information on the presence and abundance of elasmobranch prey in our study locations, thus further efforts are needed to investigate the potential relationship between elasmobranch biomass/abundance and prey availability.

Since PPAs generally cover the largest proportion of multiple-use MPAs surface area, they might play a key role as potential refuges for elasmobranch species, within MPAs, especially for those at risk of extinction, hosting different critical habitats for elasmobranchs (e.g., nursery and mating areas). Recent reviews of tagging studies showed that many elasmobranchs have pronounced site fidelity and either permanent or seasonal residency in relatively restricted areas[61,62]. In the latter case, fishing at these sites could reduce not only the limited pool of returning adult females and new-borns, but also the local abundance of individuals at older life stages. The higher BCPUE and NCPUE of elasmobranchs within PPAs found in this study also supports the idea that these areas could be acting as a potential refuge. Although our estimates are fishery-dependent, and assuming absence of hyperstability or hyperdepletion, CPUE could be considered a proxy of abundance and biomass at sea. Further studies on ecological attributes and additional investigations into the potential benefits of PPAs are needed to shed light on the ecological mechanisms that illustrate PPAs benefit for elasmobranchs.

**Improving elasmobranchs conservation**

Although different strategies to protect elasmobranchs have been implemented, their conservation status highlights the need for further efforts to halt their ongoing global decline. In the Mediterranean basin, more than half of the elasmobranch species are Critically Endangered, Endangered, or Vulnerable[47]. Many have already disappeared in several parts of the Mediterranean[52]. To date, only a few elasmobranchs (24 species), listed in Annex II of the SPA/BD protocol of the Barcelona Convention and Recommendation GFCM/42/2018/2, are fully protected in the Mediterranean Sea (they cannot be retained on board, landed, transferred, stored, sold, or displayed or offered for sale). In our study, three elasmobranch species (*Dipturus batis*, *Rhinobatos rhinobatos* and *Rostroraja alba*) listed in Annex II were caught and landed by SSF. This could be due to a lack of awareness of fishers about the status of shark and ray populations and the fishery restrictions[72]. The removal of threatened elasmobranchs, both immature and mature individuals, from inside PPAs by SSF appears to reveal a "conservation paradox". Catching immature individuals could result in a lower level of sustainable exploitation, as juveniles with delayed onset of maturity, have the greatest influence on population growth[73]. Nevertheless, the protection of mature individuals should be considered an equally important strategy for elasmobranch fisheries management, as older individuals can regularly produce young that replenish populations through consistent recruitment[74]. It is worth noting that mature individuals were also caught in UPAs, and as such from a conservation point of view, key strategies to protect elasmobranchs should focus on the protection of aggregation sites in UPAs to minimize the mortality of pregnant females during the parturition period and the establishment of a legal maximum size for captures[74]. Our findings, therefore, call for actions to establish sound SSF management strategies that ensure existing MPAs achieve their conservation goals and objectives. Additional management measures and/or stronger compliance are needed to minimize landings and by-catch mortality to halt ongoing

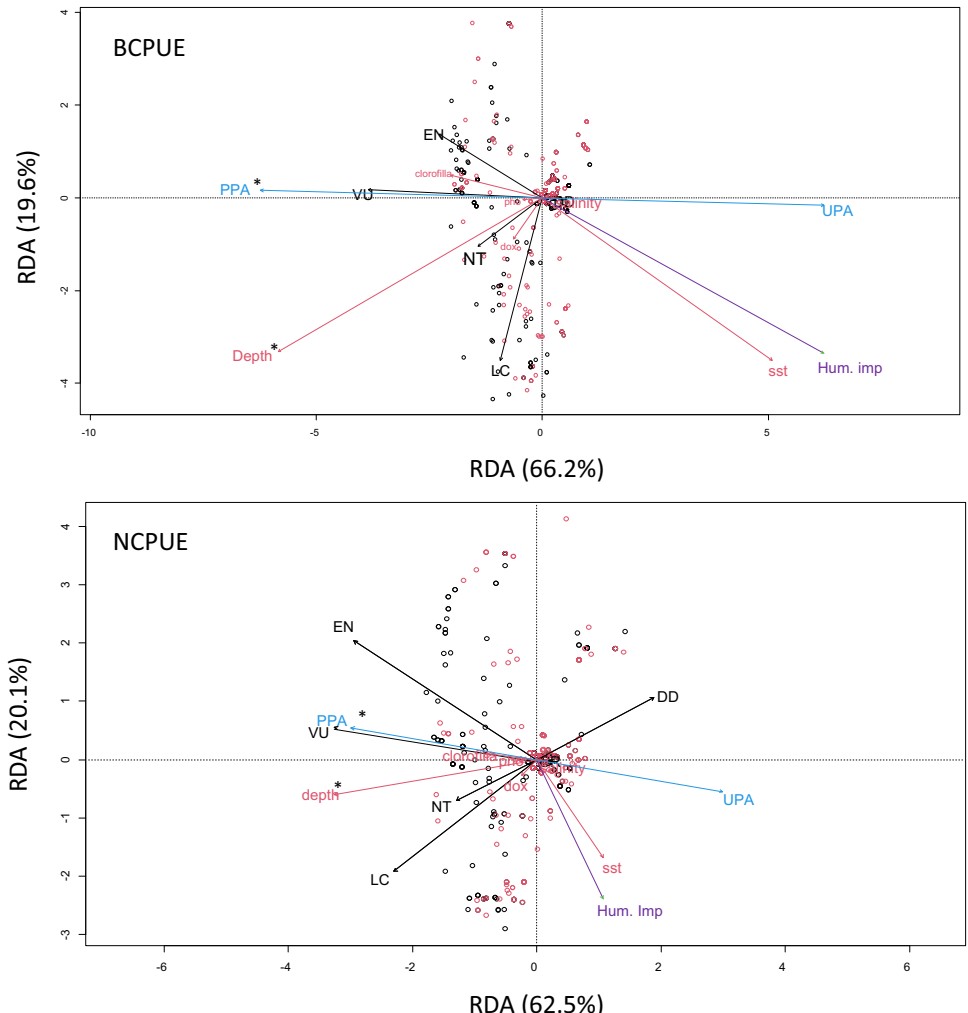

**Fig. 4 | Partial distance-based redundancy analysis (pRDA) performed on BCPUE (upper panel) and NCPUE (lower panel).** In the biplot diagrams, the species are grouped into the IUCN categories (CR = Critically Endangered; EN = Endangered; VU = Vulnerable; NT = Near Threatened; LC = Least Concern; DD = Data Deficient). Arrows indicate the correlations among variables: a longer arrow indicates that the corresponding variable is more important; a small angle between two arrows indicates that the correlation between the two variables is high. The first axis explained most of the variability ($p = 0.001$). Red circles indicate catches in PPAs black circles catches in UPAs. Significant predictors are identified with a * ($p = <0.001$). sst = sea surface temperature; dox = dissolved oxygen; pho = phosphate; Hum. Imp = human impacts.

decline and rebuild endangered elasmobranch populations[47]. In this regard, based on evidence from the literature and previous management success stories we propose four recommendations. First, fill the critical scientific data gap by finding ways to support data collection on abundance, biomass, size, spatial distribution, and habitat association of shark and batoid species, as well as life-history characteristics[75], inside and outside MPAs, coupling fishery-dependent (e.g., landings assessment, as used in the present study) and fishery-independent approaches (e.g., deploying Baited Remote Underwater Videos[5,76–78]). Second, focus on fishing-effort controls, including gear or temporal restrictions, and on the use of shark dissuasive devices[79]. Third, adopt best practices to aid the survival of by-catch species through training on the correct identification, handling, and release protocols for endangered elasmobranchs[44], and for the officials authorized to enforce regulations and check catches[12]. Fourth, improve the knowledge base, by identifying critical habitats within MPA borders that would allow for seasonal closures of nursery areas, breeding grounds and aggregation sites[63,64,80], and systematically collect SSF fishing effort data in the Mediterranean Sea that would provide more accurate estimates of total yields. The management and data collection activities needed to adhere to the four suggestions could be realised by MPAs, taking advantage of the role that MPA management authorities have as local-scale governance systems capable of implementing effective management measures. For this reason, we highlight the promising role PPAs can have, if well designed and managed, for the protection of coastal elasmobranchs.

In this study, we used the regional IUCN Red List categories[47] to guide the assessment of elasmobranch populations' response to protection. Our results showed that species listed as Threatened (i.e., Critically Endangered, Endangered, and Vulnerable) had higher B/NCPUE inside PPAs than UPAs. However, the status of the regional and national Red Lists and/or more local status of elasmobranch species may differ. Being aware of this difference is crucial for understanding more localized impacts from SSF to guide MPA management bodies in making informed decisions. IUCN Red List Categories have been found to align with the fisheries status of many exploited species and therefore provide a useful guide for prioritizing the conservation needs of different species. Species classified as Critically Endangered or Endangered cannot withstand any form of fishing[24], whereas Near Threatened or Vulnerable elasmobranchs can in some instances sustain modest levels of fishing[24]. The approach used in this study can be used to further strengthen management decisions that are much needed in areas around the world where SSF predominate (for example in North Africa[54], Mexico[22] and the Indian Ocean[60]).

In general, most countries encounter some level of difficulty, and disagreement from stakeholders, when attempting to establish new MPAs, and major capacity shortfalls hinder MPAs' potential to deliver ecological benefits[81]. Yet enhancing MPAs' coverage and ensuring their effective management is crucial if we are to achieve global conservation objectives. Many ocean experts and political actors support the proposal for a global framework to effectively protect 30% of the global ocean by 2030, known as 30 × 30, which is to be set during the next meeting of the United Nations Convention on Biological Diversity, in Kumming in 2022. The results of this study highlight that small coastal MPAs potentially offer protection to species that were previously thought not to receive benefits from MPAs. As such, multiple-use MPAs, including large and adequately managed PPAs, should improve the protection of elasmobranchs through suitable management plans that aim to reduce bycatch and promote a sustainable use of natural resources ensuring we meet conservation goals (e.g., protecting elasmobranchs and ecosystems more widely) whilst, simultaneously, allowing potentially sustainable human uses[82,83].

## Methods

### Study area
The study was conducted within and around eleven Marine Protected Areas (MPAs, *sensu lato*, including areas established under different designations) located in six EU countries of the Mediterranean Sea: Bonifacio, Cap Roux, Côte Bleue (France), Portofino, Egadi, Torre Guaceto (Italy), Es Freus, Cabo de Palos (Spain), Telascica (Croatia), Strunjan (Slovenia) and Zakynthos Island (Greece) (Fig. 1), between June 2017 and October 2018.

### Data collection
We assessed catches from 1,256 SSF operations at 11 locations (with a variable number of fishing operations per location, ranging from 37 at Telascica to 162 at Es Freus) (see Supplementary Fig. 1 for the number of fishing operations at each location). The assessed catches were from nine partially protected areas (PPAs) where fishing is allowed but regulated (see Supplementary Table 6, for details) and unprotected areas (UPAs) surrounding the 11 MPAs. Fishing location and timing for assessed catches was determined by fishers and observer availability, but attempts were made to ensure that they were spread out as much as possible over the study period. We followed the General Fisheries Commission for the Mediterranean's definition of SSF i.e., to mean fishing operations carried out by relatively small vessels, <12 meters' total length, ('length overall', LOA), that do not use towed gears[53].

The sampling activity was embedded in the framework of a larger collaborative project where small-scale fishers, MPA managers and researchers, agreed to work together to assess the drivers of effectiveness of SSF management in Mediterranean MPAs (see[84] for further details on the collaborative project). Therefore, the small-scale fishers voluntarily agreed to participate in several actions, including catch assessment. To obtain the most comprehensive dataset possible, considering that fishers have different fishing habits, we monitored catches from as many fishers as possible from among those willing to take part in the assessment (ranging from five at Torre Guaceto to twelve at Bonifacio). Catch monitoring was restricted to fixed nets as these are the most commonly used fishing gears in Mediterranean SSF. It also allowed for a reliable comparison of fishery descriptors (e.g., catch per unit of effort, CPUE) between areas. Nets were predominantly trammel nets, in about 95% of the fishing operations monitored, and the remaining were gillnets (4%) or combined trammel-gill nets (1%). These different fishing devices may be used to target different species, but they are deployed and work in the same way (i.e., they are anchored to and touch the bottom, by a lead line, and are kept in a vertical position by a float line). To obtain robust and verified data on SSF catches, we used a photo-sampling technique for catches at landing. This methodology was adopted to minimize sampling time in the field and fish manipulation, ensuring as little disturbance as possible to fishers during monitoring operations. More specifically, a scientific operator, previously trained by the project partnership, waited for the fishing vessel at landing sites, scheduling the assessment of the catch with the fisher in advance to avoid any specimens being sold before sampling. Fishers were requested to land all the catch, without throwing overboard any specimen fished. At landing, the operator spread out the catch over a flat horizontal surface and took one or more (for the largest catches) pictures to photographically capture each entire catch (thus including elasmobranchs and all other species landed), along with a ruler (as length reference) placed within the same frame. The operator ensured that each specimen was entirely visible. Each picture was associated with a unique identifier of the fishing operation (e.g., a small paper tag with a unique reference code) for the subsequent image analysis.

The type and length of fixed nets used were recorded to calculate the CPUE. Each fisher was also asked to point out on a map the approximate position and depth of net deployment. The coordinates of the fishing points were successively retrieved and used to extract data on a set of environmental variables (see below). A trained operator processed the images using the image-analysis free software ImageJ[85]. Each species was assigned to its Mediterranean IUCN category (https://www.iucnredlist.org/regions/mediterranean).

We measured the total length of each individual to the nearest 1 mm using the ruler in the picture as a reference for calibrating the measurement tool in ImageJ, and estimated the biomass (i.e., wet weight) of each specimen using specific length-weight relationships available from www.fishbase.org[86]. The number and weight of specimens of each species was used to estimate the CPUE standardized for the length of the net both in terms of abundance (NCPUE, number of individuals per 1000 m of net) and biomass (BCPUE, kg per 1000 m of net).

A size distribution was constructed for all elasmobranch species. The length at first maturity ($L_{50}$) reported in the literature (Supplementary Table 7) was used as the threshold to classify mature and immature individuals. As $L_{50}$ values of a given species may vary between males and females, and due to the impossibility to determine the sex of individuals in most photo-samples, we conservatively used the lowest between-sex value to estimate a Maturity index as $Size\ ind./L50$. This index ranges from 0 (immature) to infinite (mature), with values ≥1 indicating that the individual reached its $L_{50}$ threshold. We then calculated the proportion of mature/immature individuals grouping the elasmobranch species based on their IUCN Red list categories: Threatened (Critically Endangered, Endangered and Vulnerable: THR); Non-threatened (Near Threatened and Least Concern: NTH); Data Deficient (DD). We used a set of variables describing environmental (chlorophyll a, sea surface salinity, dissolved oxygen, phosphate, nitrate, sea surface temperature, and habitat), geographical (location, latitude, and longitude), temporal (season), bathymetric (fishing depth) and anthropogenic (mostly human impacts[87]) conditions to control for all related sources of variability and estimate the genuine effect of protection and interaction with SSF for elasmobranchs in the different fishing spots inside and outside the 11 MPAs (see Supplementary Table 1 for the list of variables considered, and their sources). The measure of overall human impacts is a compounded score including information relative to a number of human drivers of impacts (i.e., fishing, pollution, population density, climate change[87]). Features of fishing operation (i.e., fishing depth, net length, mesh size, and soak time) were also recorded and reported in Supplementary Table 6. The variables chosen were considered in the model as they had previously been identified in the literature as potential drivers of abundance, biomass and distribution of elasmobranch species (see the supplementary materials for more details about the choice of the predictors).

**Statistical analyses**

Density plots (an unbounded, continuous, and smoothed version of the histograms) were drawn to show the proportion of immature and mature individuals based on $L_{50}$ standardization of each species caught inside PPAs and UPAs. Due to the small sample size obtained for some species, density plots were drawn grouping the species based on their IUCN extinction risk categories in line with recent papers[88,89]: Threatened (THR, grouping CR, EN, and VU); Nonthreatened (NTH, grouping NT and LC); and Data Deficient (DD).

We followed a systematic approach to select the best model according to the data considered (see Supplementary Fig. 6 for a flow-chart of the key steps). To check for the presence of zero inflation in our data, we compared Poisson models vs Zero-inflated Poisson models using multiple diagnostic tools to identify the best model (i.e., Likelihood-ratio test, inspection of residuals and AIC). In case an excess of zeros was detected, to select the best-performing error family distribution, we performed the GAMLSS analysis using a "Zero inflated Poisson" (ZIP) on density and biomass of individuals (with the net length used as an offset) and compared it with "zero-adjusted Gamma" (ZAGA) on catch standardized per unit of effort. For each analysis, we used the AIC results and the graphic inspection of the residuals to select the best model. For these specific analyses, ZAGA was always found to be the best-performing modelling approach. ZAGA distribution consists of two-part model coupling, a binomial *logit*, presence/absence model (which predicts the probability of fishing operations with at least one elasmobranch individual in the catch) and a positive (truncated) abundance model (*log*, all zeroes excluded, which predicts the potential density or biomass of elasmobranchs)[90,91]. Hierarchical random models using the zero-adjusted Gamma (ZAGA) were used to model the NCPUE and BCPUE and to deal with the excess of zeros. In our case, 'location' (9 levels; Cap Roux and Cote Bleue MPAs were removed from the analyses as they do not have a PPA with fishing restrictions) and 'habitat type', which was nested in location (6 levels; see supplementary materials) were included as intercept-only random effects to account for the nested sampling design. Dissolved oxygen and nitrate were excluded from the initial set of predictors because of multi-collinearity issues (Supplementary Table 8).

The number of immature and mature individuals for each IUCN group (THR, NTH, DD) were analysed separately to assess statistical differences between PPAs and UPAs. We followed the same approach used for the above analysis implementing GAMLSS; for each combination of IUCN category and state of maturity, we also checked for the best-performing error family distribution, comparing a ZIP on counts of individuals with the net length set as offset and a ZAGA on catch data standardized per unit of effort. For each analysis, model performance was evaluated using Akaike's information criteria (AIC), with the best-fit model displaying the lowest AIC value (Supplementary Table 9). The analyses were run with 'gamlss' package for R (70). Density plots were built using the package "ggplot2"[92] in R software.

Potential outliers and excess zeros were analysed with Cleveland dotplots before fitting the models. Variance inflation factor (VIF) analysis and pairwise correlations were performed between all variables to check for collinearity[93]. Predictors with VIF > 2 were excluded from the analyses. We present the results of these tests in the supplementary information (Supplementary Table 9). We also performed the chi square test to assess if the soak time and mesh size were significantly different between PPAs and UPAs and no significant difference was detected.

Selection of the candidate predictors from each group of anthropogenic, bathymetric, geographic, temporal, and oceanographic variables (Supplementary Table 10) was performed using the StepGAIC (stepwise selection) function[94] on the full model (the model that contains all explanatory variables).

The Generalized Akaike Information Criterion (GAIC) was used to evaluate the relative goodness of fit of the candidate model set[95].

$$GAIC = 2L + kN, \tag{1}$$

where L is the log-likelihood, k is a penalty for model complexity, and N is the number of parameters in the fitted model (Supplementary Table 10). Minimization of the Schwarz Bayesian Criterion (SBC)[96], which is a particular case of the GAIC when k = log(N), was used for model comparison. A final model was selected when the SBC could not be further minimized by removing or adding terms. In order to examine the residuals for independence and identical distribution the worm plot tool was used[97]. Worm plots are detrended versions of Q-Q plots that display the discrepancy between the theoretical and empirical distributions for each observation. The structure of the distribution of the residuals indicates several features of the model fit, and these plots also include the 95% confidence interval of the unit normal quantiles[97]. After removing the effects of other factors, partial residual plots (also known as term plots) were used to look for non-linear relationships between the terms and their predictors. The "gamlss" package was used to implement all necessary functions for model fitting and evaluation, and pseudo r-squared values for each model were produced with function 'Rsq' using option 'Cragg Uhler' within the open source R 3.1.1 software[98].

Finally, a multivariate analysis was performed to investigate the effect of the protection on the BCPUE and NCPUE of the six IUCN categories. Particularly, category composition responses to protection and environmental predictors were investigated with partial redundancy analysis (pRDA), with 'location' and 'habitat' as a conditional effect using BCPUE and NCPUE of the six IUCN extinction risk categories (CR, EN, VU, LC, NT, DD). We transformed the BCPUE and NCPUE data in Hellinger distance as it has been shown to be more appropriate for data containing many zeros[99]. Random factors, 'location' and 'habitat', were included in the conditional matrix (variables to be controlled in the final models) in order to remove their effect from the analysis. The other variables were included as predictors. Based on Monte Carlo permutation with 999 iterations, the RDA was used with forward selection to filter the relative importance of explanatory variables of BCPUE and NCPUE. Statistical significance was assessed by comparing the initial F-statistic to the distribution of F-values obtained after 1,000 permutations of the response matrix and the goodness-of-fit evaluated with the adjusted $R^2$ [100]. The analyses were completed using the "vegan"[101] package within the open source R 3.1.1 software.

**Reporting summary**

Further information on research design is available in the Nature Research Reporting Summary linked to this article.

## Data availability

The data used in this study have been deposited in the public Figshare repository[102–105]. The Figshare repositories contain the data needed to reproduce the GAMLSS analyses of elasmobranch BCPUE and NCPUE and maturity between PPAs and UPAs, used in Fig. 3 and Supplementary Figs. 2–5 (https://doi.org/10.6084/m9.figshare.18318878.v1, https://doi.org/10.6084/m9.figshare.18318881.v3), as well as the multivariate analyses (partial redundancy analyses) of elasmobranch BCPUE and NCPUE between PPAs and UPAs used in Fig. 4 (https://doi.org/10.6084/m9.figshare.18318884.v1, https://doi.org/10.6084/m9.figshare.18318887.v1). Source data used in Figs. 1, 2 and Supplementary Fig. 1 are provided with this paper. Source data are provided with this paper.

## Code availability

Analyses were conducted in R and the code used to produce the results of this study is provided in R files in public Figshare repositories[106–108] (https://doi.org/10.6084/m9.figshare.18318875.v2, https://doi.org/10.

6084/m9.figshare.18318890.v1, https://doi.org/10.6084/m9.figshare.18318893.v1).

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

## Acknowledgements

We are grateful to the fishers, from all the studied fishing communities, for their invaluable contribution to this research. Many thanks are given to the MPA Directors and their staff members for their support. Habitat Information used in this publication was made available by the EMODnet Seabed Habitats project, [https://www.emodnet-seabedhabitats.eu/], funded by the European Commission Directorate General for Maritime Affairs and Fisheries. Authors are grateful to Dr. Katie E. Hogg (https://katehogg.org) for her invaluable help in reviewing the English. This research was carried out within the FishMPABlue 2 and FishMPABlue 2 Plus projects (https://fishmpablue-2.interreg-med.eu/) framework, funded by the European Territorial Cooperation Program MED and co-financed by the European Regional Development Fund (ERDF). AC was also funded by the Italian Ministry of Education, University and Research (MIUR) in the framework of the PON 'Research and Innovation 2014-2020' - section 2 'AIM: Attraction and International Mobility' (D.D.407/2018), co-financed by the European Social Fund – CUP B74I18000300001, project: 'Blue Growth' (AIM1898397 – 1).

## Author contributions

MDL, AC, ADF and PG designed the study. MDL, AC and ADF assembled data. GM performed the statistical analysis. MDL, AC, ADF and GM wrote the original draft. MDL, AC, ADF, GM, GA, CC, MM and PG revised the MS and contributed to the final version.

## Competing interests

The authors declare no competing interests.
