## [Peer Review File · Nature Communications]

Small-scale fisheries catch more threatened elasmobranchs inside partially protected areas than in unprotected areasREVIEWER COMMENTS

Reviewer #1 (Remarks to the Author):

Review of Small-scale fisheries catch more endangered elasmobranchs inside partially protected areas than in unprotected areas

The cartilaginous fish are the second most threatened major vertebrate group (after the amphibians) and are highly vulnerable to incidental overexploitation in fisheries. Data on small-scale fisheries are generally scarce, particularly at the scale seen in this study. That makes this an important study for shark and ray conservation on the Mediterranean Sea, which is itself a site of priority given the former abundance and species diversity in the area, but the high human impact over recent decades. The study appears to have been well carried out, the methods are robust and the conclusions supported by the results. I have a few queries about the methodology, but I believe these should be easily dealt with and are mostly a matter of providing more clarity and detail.

General comments

Unless I missed it (which is totally possible), the study doesn't define what they mean by a small-scale fishery. Or at least what is meant by a small-scale fishery in the context of the Mediterranean Sea. We probably all think we know what a SSF is, but this will be very context dependent on where in the world we are reading this ms. I think a clear working definition of what it means in this context would be useful.

Similar to the above, there is no explanation of what is meant by "we used a photo-sampling technique". Again, I think I know what the authors mean, but this could be spelled out to avoid confusion. Also, some detail on how the particular landing sets were selected and sampled is needed. Did the people doing the sampling wait to see a vessel coming in to dock and they photograph all of the catch? Were only elasmobranchs photographed? How did they ensure that some catch hadn't already been off-loaded etc.

Specific comments

ABSTRACT:

Line 15: I have not come across this use of "highly threatened" by the IUCN or in reference to the Red List. I understand what is meant by "threatened" with reference to the IUCN and this has an official meaning (<https://www.iucn.org/resources/conservation-tools/iucn-red-list-threatened-species>). But I do not think "highly threatened" is a term officially used by the IUCN Red List and could be confusing. I could have missed it, of course. In which case please ignore this comment. But, if I am correct, I would suggest dropping the "highly" and using the well-understood "threatened" as I think here you are referring to VU, EN and CR species. The same applies at lines 104 and 275.

Line 18: Again, it might be clearer to use threatened here as you don't just mean endangered species at this point I think. Same applies to the title -- though I appreciate that ruins the alliteration :).

INTRODUCTION:

Line 25-27: Could you specify if this figure of 800k tonnes is worldwide for the purpose of clarity?

Line 33-35: I know the paper describing the update has only just come out, but there is a chance here to update this sentence to reflect the very latest assessment. You could say "As a result, over 1/3rd of elasmobranch species are classified..." and cite Dulyv et al. 2021, Current Biology: <https://doi.org/10.1016/j.cub.2021.08.062>

Line 40-42: Again, it might be useful to refer to the new Dulyv et al. IUCN Red List paper

here – they found that 75% of threatened chondrichthyans come from coastal waters.

MATERIALS AND METHODS

Line 311-312: Here it says “see Figure 1 for the number of fishing operations at each location”, but Figure 1 does not contain that information (again, unless I missed something). Figure 1 shows the percentage of elasmobranchs caught by IUCN categories.

Line 352-353: Is something missing here – compared using the ZIP what? ZIP model, ZIP GLMM? And specify that this is the ‘pscl’ package for R?

Line 357: Martin et al. 2005 should be a numeric citation.

Line 366–369: On the random effects – if the “model formulation” in Table S8 is meant to represent the R code, then it looks like this was a hierarchical random effect (habitat type was nested in location), which makes sense given what you describe as the nested sampling design. Can you make this clear that you used a nested random effect structure.

Line 374: VIF results seem to be in Table S7, not Table S5.

Line 378-379: I am not sure what is meant by scale and shape parameters settled constant – can you try rephrasing this? And I think this should say something like “The relative goodness of fit of the candidate model set was assessed using the...”. As AIC and its derivatives are measures of relative goodness of fit (not absolute).

Line 382: I’ve not heard k called the overfit penalty parameter before – perhaps rephrase as “the penalty for model complexity” or the “parameter that penalises overfitting”.

Line 406: 1,000 rather than 1.000?

Line 408: Be consistent with how R is cited? See line 395.

RESULTS AND DISCUSSION

Line 122 and 123: Is gr. meant to be grams? If so, use g as the SI abbreviation? Also, would it make sense to standardise the presentation of BCPUE into either g or kg? It is in kg in figure 2 and g in the text.

Line 133-136: I think this needs a rephrase. You say that immature specimens from the high risk groups were caught most frequently – but (unless I misunderstood it) the figure shows that mature individuals from the LR group were the most commonly occurring. What I think you mean is that “immatures individuals were caught more frequently than mature individuals for species in the HR and DD categories”.

Lines 180-182: Check the text here properly reflects the results. It says that “immature individuals of HR and DD were more frequently captured inside PPAs” but then the statistics give p-values less than 0.05 for all three categories. However, looking at the means and the spread of the SEs for immatures in the DD group it doesn’t look like that difference would be statistically significant on the face of it?

Lines 189-190: Are these means calculated from the raw data? If so, how meaningful are they given all the zeros in the data? Would it be possible to present modelled means? Also, the values here for BCPUE are ~ 1 kg/1000 m of net. But looking at the model output in Table S5 the exponential of the intercept (which should be the mean for one of the PPA categories) is $\exp(8.6972) = 5986$ – was the analysis done on the data in grams?

FIGURES AND TABLES:

Figure 1 legend: Very minor point – and you might have to abide by journal style, but the technically correct usage (as I understand it) when using the IUCN categories is to use capital letters. i.e. Critically Endangered etc.

SUPPLEMENTARY MATERIALS

Line 10: This might sound like a daft comment, but could you clarify what you mean by season as one of the predictors. Your study was conducted from July to October, so is this simply boreal summer (July and August) and boreal autumn (September and October)?

Line 12: Fishers rather than fishermen (as in the main text)? And they provided the “approximate depth”, at least according to the main text.

Line 14: What is the difference between sandy mud and muddy sand?

Line 22-25: The results of the correlations between the predictors is in Table S7, not Table S6. And Table S3 shows mean catches in each of the MPAs. So doesn't seem to be relevant here. Also, I'd suggest writing out the abbreviations (VIF, GAMLSS and ZAGA) here so that readers don't have to flick back to the main text if they are not familiar with these names.

Reviewer #2 (Remarks to the Author):

Title

Small-scale fisheries catch more endangered elasmobranchs inside partially protected areas than in unprotected areas

Note: I added line numbers for easier referencing

The authors have presented excellent and novel research illuminating catch composition and distribution of elasmobranchs in a hotspot of elasmobranch threat. Understanding the catches and composition is of clear conservation importance and the authors found that at some of the locations (as shown on Figure 1) the percent of threatened species is >50% of the catch. I think the paper could be strengthened by clearly answering the questions the authors set out in their introduction. I think there is some methods and discussion that is not really important to the main point of the paper (fully protected versus PPAs), I would be careful about overstating the importance of PPAs without discussion of some of the factors that may influence the higher catch estimates in the PPAs versus outside of PPAs and catches do not necessarily translate to abundance. I suggest emphasizing the catch composition at the sites as beyond Figure 1 it is not discussed (could discuss the number of sites with >50% of catch comprised of threatened species, etc). Also, a discussion if the catches from PPAs in compliance with current regulations could be useful to discuss.

Suggestions

I think your short title could be more informative. Too many acryoyms. Something like: Elasmobranch catches inside and outside partially protected areas

Abstract

There is no IUCN category of 'highly threatened' there is the threatened categories of Critically Endangered, Endangered, or Vulnerable.

Introduction

Small suggestions: type out numbers under 10.

Starting line 52: it could be worth using the terminology that is used by the MPA literature as that could keep the text consistent and also cut down on some of the acronyms. Marine Reserves are those MPAs that are strictly protected and prohibit fishing. A MPA is therefore those MPAs that are partially protected or multiple use, etc. So Marine Reserves have been shown to have positive effects for sharks, however, MPAs the evidence on biomass and abundance changes is less clear.

Line 40: instead of sharks and batoids just say elasmobranchs

Line 80 missing end bracket

Line 82: you discuss fully-protected and ppas AND 'their potential to protect elasmobranch species' but this is not carried throughout the text. You could discuss the catch and composition differences between the two types of MPAs

Line 86: "Due to the likely higher fishing pressure in UPA than in PPAs, we hypothesize that elasmobranch catch per unit of effort (CPUE) is higher in PPAs than UPAs." since you found this to be true – how do you reconcile your results with this?

Line 84 – make the covariates section it's own question. What are you asking here?

Methods

Line 315. You say fishing inside the PPAs is regulated – how? And would this affect the fishers tell you their catch was from inside a PPA?

Line 342: "estimate the genuine effect of protection and the SSF interaction with elasmobranchs in the different fishing spots inside and outside the 11 MPAs" Can you clarify what you are asking with this third analysis.

The authors clearly have an excellent grasp on statistical analyses

Results

Line 112: IUCN categories are capitalized – Critically Endangered, Endangered, etc.

Figure 1 – you can make the text much bigger. IUCN categories are capitalized. You could include a bigger map to identify more broadly the country.

Line 136 to 143 - this could be in your discussion as it is not a result. "this result might indicate..."

Line 158 – this is good discussion, I suggest moving down

Line 162 to 176 – this is also good discussion. Consider moving down and have the results start again at line 178 "Higher elasmobranch catches per unit..."

Figure 2 – this figure is great, however, I suggest making the text as big as you can as it's hard to see. You could split up the barplot up by IUCN category as it would be interesting to see the difference between the threatened species (CR, EN, VU).

Rhinobatos Rhinobatos should not have a capital letter. You could potentially bold the species that have a high number of individuals caught within versus outside of MPAs.

Figure 3 – same suggestion. Make text bigger

L50 versus L50 choose one and make consistent through text

Line 218 – I don't think you can say the restrictions on human use lead to a "increase' of density and biomass as you haven't showed that through time. Rather you can say it is correlated with. Were these MPAs placed there because they are important to sharks? Could fisheries be targeting sharks within MPAs and would that influence you CPUE estimates?

I suggest being cautious about saying 'abundance' in place of CPUE (biomass or abundance). Although a very useful way to infer abundance the catch statistics are not fisheries independent estimates of 'true' abundance.

Discussion

You introduce fully protected MPAs and PPAs in Figure 1 but don' discuss it again. What I think is interesting is that the fully protected MPAs have LC species but no VU, EN, or CR species. These MPAs are the ones that would lead to an increase of abundance and

biomass whereas the PPAs are the ones with high threatened catches. Implementing regulations in the PPAs with high shark catches could be a conservation win.

I think a paragraph discussing how the catch statistics were collected and how that could influence your results would be useful. Are fishers targeting sharks in PPAs versus UPAs? Are these PPAs established to protect sharks?

Reviewer #3 (Remarks to the Author):

General comments

The study describes the assessment of elasmobranch catch in small-scale fisheries in the Mediterranean, comparing catch rates in areas of differing protection status. The study consists of a comprehensive dataset spanning 11 locations within six countries. I think the premise of the paper is sound and on a subject matter in need of attention, especially in an area such as the Mediterranean. I do feel however, the description and presentation of methods and results are lacking and do not provide enough evidence to substantiate the main conclusions. I think the lack of detail on the fisheries themselves is a major issue when comparing catch rates. There is little information on whether the fishers differ across countries, the gears being deployed, and key elements of effort such as soak time, and number of sets – which will hugely influence standardising and comparing catch rates.

- I don't feel the methods are fully described, especially with regards data collection or assessment of the fishery itself.
- I'm not sure the data presented necessarily back up conclusions and statements made.
- I don't feel the study is novel enough for Nature Communications.
- The absence of some key features of catch effort are very important to analysis.
- Presentation and explanation of figures is limited.
- Little information on what constitutes a PPA – what are the restrictions? Do they differ across countries?
- MPAs are mentioned but not compared. I understand there are fewer, but may have been interesting to see if there is a scale from no protection to full protection in terms of catch.
- The abstract headlines are that more elasmobranchs were caught in PPAs compared to UPAs, and that this means protection can have benefits but SSF are risk to conservation. I see the point the authors are making, but I think that the fact that <1 shark per operation is being caught, which is very low, suggests intensive fishing likely already depleted many populations. I would also argue that the main finding lies in the higher number of immature individuals being caught in PPAs – have these PPAs been placed in nursery grounds?

Specific comments

Title – Endangered should be changed to threatened as the descriptions used grouped IUCN classifications.

Line 15 – “Highly threatened” is used, what is this classification, it is not used elsewhere. Is this the same as the IUCN classification for threatened, and what the authors declare “high risk” later in the manuscript?

Lines 27-29 - This is not the only reason. SSF, which are under reported and are a source of huge shark catch.

Line 40 - Not just offshore. Reported catch of elasmobranchs largely originates from industrial fleets, however much of this occurs in coastal waters on the continental shelf.

Lines 57-58 - Global estimates are between 5-7.5% of the ocean being in protected areas, below 10% target and way off new target of 30%. How much is 29% of 5%, and how much area. Should note that there are no shark sanctuaries in the Atlantic outside the Caribbean Sea.

Line 88 - Need more detail on the gear. What type of fixed net, mesh size, how is it fixed i.e. demersal set or set from the surface. What is the target species for this fishery?

Line 96 - I would caution against claiming SSF to be the threat to elasmobranchs in the Mediterranean. These fisheries are catching quite low levels relatively based on the results and so SSF may exacerbate the situation or catch more immature individuals, but these comparisons are not presented here.

Figure 1 - Are these locations where the catches were landed or fished (or both) as these are describing PPAs, and MPAs but not UPAs. Where is the composition of UPA catch?

- The caption states "i.e. Biomass CPUE", this is confusing, are the data from BCPUE? As these may differ from composition by individuals.

Line 105 - 1256 operations carried out - What constitutes an operation? How many sets were deployed per operation? Across what timeframe were these data collected? Soak time? Number of hours/days? Does time of day fishing change? Is <1 shark per operation a lot?

Figure 2 - In my opinion this figure doesn't work. I can see what the authors are trying to do to make an interesting plot, but I think a simpler presentation of the results would be more informative. Also the colours are not very distinguishable to colour-blindness.

Line 134 - I'm not sure why the authors have redefined a category established by the IUCN. Species in VU, EN, and CR are classified collectively as "threatened".

Lines 145-146 - Is this a little contradictory as many of the HR species are demersal too?

Figure 3 - This figure needs more explanation. I think the two smaller graphs should be bigger and aligned to the right of the main panel, so that they are half the height of the main figure.

Lines 185-186 - This statement only applies to HR and DD species - I think this should be made more explicit.

Line 192 - I think using a word like confirmed is a little strong when presenting these results.

Lines 192-193 - Why are test results not presented here?

Lines 195-197 - I assume the relationships described are non-significant? Can the test statistics be presented here?

Line 214 - Is density the right description? Is it occurrence or presence maybe?

Lines 241-251 - I think this paragraph is too broad and too vague, I think it needs to be incorporated with the next section where examples are given for strategies rather than simply stating conservation needs to be done.

Line 243 – “Half of the elasmobranchs are Critically Endangered” should be “half the elasmobranch species are Critically Endangered”.

Line 275 – Consistency of terms – “highly threatened” has not been defined, either refer to IUCN classifications or the High Risk category defined and used throughout.

Line 310 – Again details on the fishery and fishing operation are needed.

Line 312 – More information required for photo-sampling technique. Were individuals photographed separately, or more general pictures taken to count and identify later?

Line 316 – What were the types and lengths of the nets? Also are there other details on the fishery gathered – soak time? Number of sets? Duration?

Line 326-328 – Are these done nominally? To properly model a rate, an offset should be used to account for variation in a defined effort metric.

Line 338 – What resolution were environmental variables acquired at? Any standardising? Would daily rasters have given a better account of environmental conditions?

Line 340 – Why was bathymetric depth used and not depth of gear?

Line 340 – What is the layer of human pressure?

Lines 352-353 – I’m not sure what this model is referring to. The length data described is of proportions of mature and immature individuals. What is the zero-inflated count of?

Line 364 – I would again argue the best approach would be to use an offset to model the rate of CPUE. Type of net was said to have been asked when interviewing the fishers, is this not included as a variable in the models?

Responses to reviewers

Reviewer #1:

The cartilaginous fish are the second most threatened major vertebrate group (after the amphibians) and are highly vulnerable to incidental overexploitation in fisheries. Data on small-scale fisheries are generally scarce, particularly at the scale seen in this study. That makes this an important study for shark and ray conservation on the Mediterranean Sea, which is itself a site of priority given the former abundance and species diversity in the area, but the high human impact over recent decades. The study appears to have been well carried out, the methods are robust and the conclusions supported by the results. I have a few queries about the methodology, but I believe these should be easily dealt with and are mostly a matter of providing more clarity and detail.

We thank the reviewer for appreciating our work. We reworked the MS to comply with her/his suggestions (as detailed below), which we believe have helped us significantly improve MS's clarity and readability.

General comments

Unless I missed it (which is totally possible), the study doesn't define what they mean by a small-scale fishery. Or at least what is meant by a small-scale fishery in the context of the Mediterranean Sea. We probably all think we know what a SSF is, but this will be very context dependent on where in the world we are reading this ms. I think a clear working definition of what it means in this context would be useful.

We thank the reviewer for this useful comment, particularly relevant considering the inconsistency in the use of the term SSF and the lack of strict legal definition in a number of national legislations. We have better defined, at the beginning of the 'Data collection' paragraph, what are generally considered SSFs in the context of the Mediterranean Sea, following the operational definition adopted by the General Fisheries Commission for the Mediterranean of the FAO (GFCM), in the recent report "The State of Mediterranean and Black Sea Fisheries, 2020". Specifically, we defined as SSF those fishing operations carried out by vessels <12 meters LOA ('length overall'), and not using towed gears. This definition is in line with the one very recently adopted by the European Parliament (Regulation (EU) 2021/1139 of the European Parliament and of the Council of 7 July 2021 establishing the European Maritime, Fisheries and Aquaculture Fund and amending Regulation (EU) 2017/1004)

Similar to the above, there is no explanation of what is meant by "we used a photo-sampling technique". Again, I think I know what the authors mean, but this could be spelled out to avoid confusion. Also, some detail on how the particular landing sets were selected and sampled is needed. Did the people doing the sampling wait to see a vessel coming in to dock and they photograph all of the catch? Were only elasmobranchs photographed? How did they ensure that some catch hadn't already been off-loaded etc.

We thank the reviewer for this comment, and we apologize for the lack of detail in the previous version of the MS. We developed the section 'Data collection' adding all the relevant information about the photo-sampling methodology requested by the referee. Concerning the sampling setting, it is relevant to point out that the sampling activity was embedded in the framework of a larger collaborative project where small-scale fishers, together with marine protected areas' managers and researchers, agreed to collaborate to assess the drivers of effectiveness of SSF management in Mediterranean MPAs (see Di Franco et al. 2020 for further details on the collaborative project). Therefore, the small-scale fishers voluntarily agreed to participate in a number of actions, including the assessment of the catches. At each location, a scientific operator previously trained by the project partnership, waited for the fishing vessel at the port, scheduling the assessment of the catch with the fisher in advance, in order to intercept the vessel at its arrival at the port, and to avoid any specimens being sold before the sampling. Fishers were previously instructed to land all the catch, without throwing overboard any specimen fished. At landing the operator spread out the catch over a flat horizontal surface and took one or more (for the largest catches) pictures to photographically capture each entire catch (thus including elasmobranchs and all other species landed), along with a ruler (as length reference) placed within the same frame. The operator ensured that each specimen was entirely visible. Each picture was

associated with a unique identifier of the fishing operation (e.g. a small paper tag with a unique reference code) for the subsequent image analysis. The above details are now specified in the revised version of the MS. We adopted sampling at landing, rather than having an operator on board, because, of size limitations of SSF vessels and for safety reasons imposed by national laws that generally only allow registered fishers to be on-board. Securing an operator on board for all the fishing operations would have not been possible. As such, we cannot ensure that all the catch was landed and some specimens were not discarded before landing., However due the fishers' engagement in the wider participatory framework described above and trust built between the project partners and fishers (who were given instructions to land the entire catch), we have confidence that no specimens were thrown overboard. Please, see lines 419-445 in the clean version.

Di Franco A., Hogg K.E., Calò A., Bennett N.J., Sévin-Allouet M.A., Esparza A.O., Lang M., Koutsoubas D., Prvan M., Santarossa L., Niccolini F., Milazzo M., Guidetti P. (2020). Improving marine protected area governance through collaboration and co-production. Journal of Environmental Management, 269, 110757. <http://dx.doi.org/10.1016/j.jenvman.2020.110757>

Specific comments

ABSTRACT:

Line 15: I have not come across this use of “highly threatened” by the IUCN or in reference to the Red List. I understand what is meant by “threatened” with reference to the IUCN and this has an official meaning (<https://www.iucn.org/resources/conservation-tools/iucn-red-list-threatened-species>). But I do not think “highly threatened” is a term officially used by the IUCN Red List and could be confusing. I could have missed it, of course. In which case please ignore this comment. But, if I am correct, I would suggest dropping the “highly” and using the well-understood “threatened” as I think here you are referring to VU, EN and CR species. The same applies at lines 104 and 275.

We thank the reviewer for highlighting this point, also raised by the other reviewers. We totally agree with the referee that the term used is confusing, and we changed it with “threatened” throughout the entire MS.

Line 18: Again, it might be clearer to use threatened here as you don't just mean endangered species at this point I think. Same applies to the title -- though I appreciate that ruins the alliteration :).

Done, as requested. We revised the sentence and the title accordingly.

INTRODUCTION:

Line 25-27: Could you specify if this figure of 800k tonnes is worldwide for the purpose of clarity?

Done as requested. This datum is indeed referred to worldwide catches.

Line 33-35: I know the paper describing the update has only just come out, but there is a chance here to update this sentence to reflect the very latest assessment. You could say “As a result, over 1/3rd of elasmobranch species are classified...” and cite Dulyv et al. 2021, Current Biology:

<https://doi.org/10.1016/j.cub.2021.08.062>

We thank the reviewer for suggesting this very useful reference. The suggested paper, in fact, had not been published yet when we submitted our MS to the journal. We revised the sentence and cited the paper according to referee's suggestion.

Line 40-42: Again, it might be useful to refer to the new Dulvy et al. IUCN Red List paper here – they found that 75% of threatened chondrichthyans come from coastal waters.

Done, Thank you. We added the suggested reference.

MATERIALS AND METHODS

Line 311-312: Here it says “see Figure 1 for the number of fishing operations at each location”, but Figure 1 does not contain that information (again, unless I missed something). Figure 1 shows the percentage of elasmobranchs caught by IUCN categories.

We thank the referee for spotting this, and we apologize for the wrong reference in the previous version of the manuscript. Following this comment, we included a new figure (Fig. S1, supplementary material) showing the number of fishing operations we monitored at each location, inside and outside the MPAs. This is now cited in the main text of the revised MS.

Line 352-353: Is something missing here – compared using the ZIP what? ZIP model, ZIP GLMM? And specify that this is the ‘pscl’ package for R?

We thank the referee for spotting this. Based also on a suggestion provided by the referee3#, we re-run some analyses and modified the sentence. We performed GAMLSS models on number of individuals using an error family distribution of Zero-inflated Poisson (ZIP) with the net length used as an offset. This is now specified in the revised version of the MS.

Line 357: Martin et al. 2005 should be a numeric citation.

We thank the referee for spotting this. This has been corrected.

Line 366–369: On the random effects – if the “model formulation” in Table S8 is meant to represent the R code, then it looks like this was a hierarchical random effect (habitat type was nested in location), which makes sense given what you describe as the nested sampling design. Can you make this clear that you used a nested random effect structure.

We thank the referee for this very pertinent comment. We revised the text to clarify that we used a hierarchical random effect model.

Line 374: VIF results seem to be in Table S7, not Table S5.

We thank the referee for spotting this. We revised the text to refer to the right supplementary table.

Line 378-379: I am not sure what is meant by scale and shape parameters settled constant – can you try rephrasing this? And I think this should say something like “The relative goodness of fit of the candidate model set was assessed using the...”. As AIC and its derivatives are measures of relative goodness of fit (not absolute).

We apologize for the lack of clarity. We acknowledge this phrasing was confusing and probably misleading. We reworked the entire paragraph to enhance clarity and we also changed the sentence about the AIC as suggested.

Line 382: I’ve not heard k called the overfit penalty parameter before – perhaps rephrase as “the penalty for model complexity” or the “parameter that penalises overfitting”.

Thanks for the useful suggestion, we modified the sentence as requested.

Line 406: 1,000 rather than 1.000?

We thank the referee for spotting this. We revised the text as suggested.

Line 408: Be consistent with how R is cited? See line 395.

We thank the referee for spotting this. We revised the citation to ensure consistency throughout the text.

RESULTS AND DISCUSSION

Line 122 and 123: Is gr. meant to be grams? If so, use g as the SI abbreviation? Also, would it make sense to standardise the presentation of BCPUE into either g or kg? It is in kg in figure 2 and g in the text.

We thank the referee for this useful comment. We replaced gr with g. As suggested, we also standardised all values to grams both in the text and the figures.

Line 133-136: I think this needs a rephrase. You say that immature specimens from the high risk groups were caught most frequently – but (unless I misunderstood it) the figure shows that mature individuals from the LR group were the most commonly occurring. What I think you mean is that “immatures individuals were caught more frequently than mature individuals for species in the HR and DD categories”.

We thank the referee for this useful comment. We rephrased as suggested.

Lines 180-182: Check the text here properly reflects the results. It says that “immature individuals of HR and DD were more frequently captured inside PPAs” but then the statistics give p-values less than 0.05 for all three categories. However, looking at the means and the spread of the SEs for immatures in the DD group it doesn't look like that difference would be statistically significant on the face of it?

Thanks for this comment. As reported, based also on comments by referee 3#, above we re-run the analyses to assess differences of mature and immature individuals between PPA and UPA using a different error distribution. We added the new analyses and modified the text that now reads “The analyses of the proportion of mature and immature elasmobranchs did not find any statistical difference between PPA and UPA for mature individuals of TH, LR and DD groups (TH, $Z_{39} = 0.946$ $p = 0.998$; LR, $Z_{91} = -1.242$ $p = 0.217$; DD, $Z_{32} = 0.432$ $p = 0.669$; Figure S2). On the contrary, the statistical analyses revealed that immature individuals of TH and DD species were more frequently captured inside PPAs (TH, $Z_{85} = -3.268$ $p = 0.0015$; DD, $Z_{29} = -2.006$ $p = 0.006$; Figure S2), whilst for the LR group no statistical differences were found between protection levels ($Z_{86} = 0.748$ $p = 0.456$).”. Please, see lines 216-221 of the clean version.

Lines 189-190: Are these means calculated from the raw data? If so, how meaningful are they given all the zeros in the data? Would it be possible to present modelled means? Also, the values here for BCPUE are ~1 kg/1000 m of net. But looking at the model output in Table S5 the exponential of the intercept (which should be the mean for one of the PPA categories) is $\exp(8.6972) = 5986$ – was the analysis done on the data in grams?

We thank the referee for spotting this, and we apologize for the lack of consistency in the previous version of the MS. This has now been fixed in the revised version of the MS. We analysed the data in grams and due to the large proportion of zeros in our dataset we decided to use an appropriate model for this kind of data (zero inflated model). Concerning the Figure 2 we prefer to use the raw data because they represent the reality rather than using a prediction. As requested, we converted all the values in grams in both figures and text.

FIGURES AND TABLES:

Figure 1 legend: Very minor point – and you might have to abide by journal style, but the technically correct usage (as I understand it) when using the IUCN categories is to use capital letters. i.e. Critically Endangered etc.

We thank the referee for this suggestion, and we revised the text accordingly.

SUPPLEMENTARY MATERIALS

Line 10: This might sound like a daft comment, but could you clarify what you mean by season as one of the predictors. Your study was conducted from July to October, so is this simply boreal summer (July and August) and boreal autumn (September and October)?

We apologize for the lack of clarity. We have clarified in the MS that the study was carried out along 16 months from June 2017 to October 2018 and season predictor included four levels: “Winter”, “Spring”, “Summer”, “Autumn”. Each fishing operation was assigned to a specific boreal season based on its calendar date.

Line 12: Fishers rather than fishermen (as in the main text)? And they provided the “approximate depth”, at least according to the main text.

Thank you for spotting this. We revised the text accordingly.

Line 14: What is the difference between sandy mud and muddy sand?

These are technical categories of substrata used in the Eunis habitat classification (<https://eunis.eea.europa.eu/>) and based on Folk's geological classification. The both are mixed sediments composed by different fractions of sediment types with variable grain size. A muddy sand is mainly formed by sands (fine or very fine-grained) with a low, but relevant, fraction of silt. On the contrary, a sandy mud is a type of mud with a significant fine to very fine sand fraction. In the revised version of the MS we have now made explicit reference to Eunis classification and referenced the appropriate website.

Line 22-25: The results of the correlations between the predictors is in Table S7, not Table S6. And Table S3 shows mean catches in each of the MPAs. So doesn't seem to be relevant here. Also, I'd suggest writing out the abbreviations (VIF, GAMLSS and ZAGA) here so that readers don't have to flick back to the main text if they are not familiar with these names.

We thank the referee for spotting this. We revised the text accordingly.

Reviewer #2 (Remarks to the Author):

Title

Small-scale fisheries catch more endangered elasmobranchs inside partially protected areas than in unprotected areas

Note: I added line numbers for easier referencing

The authors have presented excellent and novel research illuminating catch composition and distribution of elasmobranchs in a hotspot of elasmobranch threat. Understanding the catches and composition is of clear conservation importance and the authors found that at some of the locations (as shown on Figure 1) the percent of threatened species is >50% of the catch. I think the paper could be strengthened by clearly answering the questions the authors set out in their introduction. I think there is some methods and discussion that is not really important to the main point of the paper (fully protected versus PPAs), I would be careful about overstating the importance of PPAs without discussion of some of the factors that may influence the higher catch estimates in the PPAs versus outside of PPAs and catches do not necessarily translate to abundance. I suggest emphasizing the catch composition at the sites as beyond Figure 1 it is not discussed (could discuss the number of sites with >50% of catch comprised of threatened species, etc). Also, a discussion if the catches from PPAs in compliance with current regulations could be useful to discuss.

We thank the referee for appreciating our work and for the useful comments provided. As requested, in the first paragraph of the discussion section we added a text discussing on catch composition, as showed in Figure 1. Please, see lines 142-150.

For what concerns the other point, currently, species-specific measures (e.g full ban on catching, storing, landing, trans-shipping, and selling) are in place in the Mediterranean Sea for very few elasmobranchs. Three (*Dipturus batis*, *Rhinobatos rhinobatos* and *Rostroraja alba*) of the 24 species caught by SSF during our study, are listed in the Annex II of the SPA/BD Protocol and fishers could have landed those because of the lack of awareness about the fishery restrictions for the three species; we discussed this aspect in the conclusion section; please, see lines 258-267 of the clean version.

To the best of our knowledge, there are no MPAs specifically established for elasmobranchs in the Mediterranean and Black Sea and the PPAs included in our study do not have any specific restrictions for elasmobranch catches.

Suggestions

I think your short title could be more informative. Too many acronyms. Something like:

Elasmobranch catches inside and outside partially protected areas

Thank you for this suggestion. The short title has been revised accordingly.

Abstract

There is no IUCN category of ‘highly threatened’ there is the threatened categories of Critically Endangered, Endangered, or Vulnerable.

Thank you for this point also raised by the other reviewers. We modified the text, using “threatened “, to ensure alignment with IUCN categories.

Introduction

Small suggestions: type out numbers under 10.

Thank you for the suggestion. We typed out the numbers under 10.

Starting line 52: it could be worth using the terminology that is used by the MPA literature as that could keep the text consistent and also cut down on some of the acronyms. Marine Reserves are those MPAs that are strictly protected and prohibit fishing. A MPA is therefore those MPAs that are partially protected or multiple use, etc. So Marine Reserves have been shown to have positive effects for sharks, however, MPAs the evidence on biomass and abundance changes is less clear.

We thank the referee for this very pertinent and useful comment. To clarify the use of the rich and sometimes confusing terminology about MPAs, we decided to refer to a recently published paper in *Science* (Gorrud-

Colvert et al, 2021), not yet published when we submitted the MS to the journal. The term Marine Protected Area (MPA) indicates an array of area-based tools where conservation is the primary goal. MPAs are defined based on different criteria, among them the ‘protection level’, which may range from minimal to full protection (this latter corresponding to marine areas where all types of extraction activities are forbidden). A fully protected MPA thus corresponds to what was called in the past ‘Marine Reserve’. Multiple-use MPAs, depending on the levels of protection within subzones, may include both fully protected areas and partially protected areas (these latter corresponding to areas where protection is ‘high, light or minimal’, based on Grorud-Colvert et al, 2021), where human activities, including fishing, are allowed but more or less restricted. Therefore, following the terminology provided in this recent paper, we used only the term “fully protected areas” to refer to those areas where all extractive activities are banned and the term ‘Partially protected area’ to indicate MPAs where fishing activities are allowed but regulated. We revised the text accordingly, ensuring consistency throughout the entire MS and citing the paper Grorud-Colvert et al, 2021.

Line 40: instead of sharks and batoids just say elasmobranchs
We revised the text as suggested.

80 missing end bracket
We thank the referee for spotting this. We revised as suggested.

Line 82: you discuss fully-protected and ppas AND ‘their potential to protect elasmobranch species’ but this is not carried throughout the text. You could discuss the catch and composition differences between the two types of MPAs

We thank the referee for this comment. We apologize if in the previous version of the MS we may not have been clear enough about our sampling design. We compared the catches composition of commercial fishing operations occurred within the PPAs with operations carried out in unprotected areas (UPAs) outside the MPAs. We did not collect data from Fully Protected Areas (FPAs) as all fishing activities are banned in those zones. Therefore, we were able to monitor catches from PPAs and unprotected areas. For this reason, as Cap Roux and Cote Bleue include only FPAs, we collected the data only in UPAs around the two MPAs. As a result, these two MPAs were removed from the statistical analyses in which we tested the potential effect of protection by comparing catches from PPAs with the ones from UPAs. We modified the sentence to better explain that. In addition, we created a new figure showing the number of fishing operations we monitored at each location in both PPAs and UPAs.

Line 86: “Due to the likely higher fishing pressure in UPA than in PPAs, we hypothesize that elasmobranch catch per unit of effort (CPUE) is higher in PPAs than UPAs.” since you found this to be true – how do you reconcile your results with this?

Thanks for the comment. This part is discussed in the results and discussion section, and now reads “our results suggest that PPAs may play an important role in protecting threatened elasmobranch species along the Mediterranean coast. This finding implies that restrictions to human use as found in PPAs (reduced fishing effort, the use of less-impacting fishing gears, and the reduction of other sources of human disturbance in general) could be correlated to an increase of density and biomass of these threatened species” lines 232-237

Line 84 – make the covariates section it’s own question. What are you asking here?

Thanks for this useful comment. We modified the sentence accordingly, and it now reads “we used a set of covariates to disentangle the effect of these variables in order to see the real effect of protection variable”.

Methods

Line 315. You say fishing inside the PPAs is regulated – how? And would this affect the fishers tell you their catch was from inside a PPA?

We thank the referee for this comment. We added a table (Table S6 in Supplementary Materials) showing the restrictions to SSF in each of the PPAs included in our study. Excluding the 2 MPAs only covered by FPAs (Cote Bleue and Cap Roux) where fishing is totally prohibited, in fact, all other PPAs encompass one or more fishing restrictions/regulations compared to the surrounding UPAs including: limited entry, gear restrictions (e.g. mesh size or net length regulations), time restrictions (e.g. the use of certain gears only in specific seasons), territorial use rights for fishers and/or fish size limits. Concerning the second point raised by the reviewer, as reported in response to a comment raised by referee 1, the fishers were engaged in a wider

participatory project and were fully supportive of the activities carried out. This, together with the fact that the fishers were authorized to fish inside the PPAs and would not have had any evident advantage to misreport their catches, bring us to be relatively confident that fishers did not provide false information on the provenience of their catches.

Line 342: “estimate the genuine effect of protection and the SSF interaction with elasmobranchs in the different fishing spots inside and outside the 11 MPAs” Can you clarify what you are asking with this third analysis. The authors clearly have an excellent grasp on statistical analyses

We thank the referee for this comment, and we apologize for the lack of clarity in the previous version of the MS. Referring to “the genuine effect” we wanted to clarify that a set of variables describing environmental (chlorophyll a, sea surface salinity, dissolved oxygen, phosphate, nitrate, sea surface temperature and habitat), geographical (location, latitude and longitude), temporal (season), bathymetric (depth of the net deployed in the water) and anthropogenic (human pressure) features were used to control for the variability associated to these elements, other than the one relative to protection, and disentangle the effect related to the presence of protection measures (i.e. within the PPAs). We rephrased the text to clarify this point. Please, see lines 94-95

Results

Line 112: IUCN categories are capitalized – Critically Endangered, Endangered, etc.

We thank the referee for this comment, shared also by referee 1. We revised the text accordingly.

Figure 1 – you can make the text much bigger. IUCN categories are capitalized. You could include a bigger map to identify more broadly the country.

Thanks for your suggestion. We modified the Fig. 1 as requested

Line 136 to 143 - this could be in your discussion as it is not a result. “this result might indicate...”

Thank you for the suggestion. In our MS we opted to merge the results and discussion section, and divided them in 3 chapters (“SSF mostly caught highly threatened species”, “Higher elasmobranch catches per unit of effort inside partially protected areas” and “Improving elasmobranchs conservation”). These lines represent a discussion of the first chapter.

Line 158 – this is good discussion, I suggest moving down

We thank the referee for the suggestion, however, as specified for the previous comment we opted for merging results and discussion section, and we believe these lines fit the discussion related to the overall pattern of catches of elasmobranchs by SSF in the Mediterranean Sea.

Line 162 to 176 – this is also good discussion. Consider moving down and have the results start again at line 178 “Higher elasmobranch catches per unit...”

Also in this case, we thank the referee for the suggestion, but we think that this specific part of the discussion fits better the first chapter of results/discussion section about the interaction of elasmobranchs with SSF.

Figure 2 – this figure is great, however, I suggest making the text as big as you can as it’s hard to see. You could split up the barplot up by IUCN category as it would be interesting to see the difference between the threatened species (CR, EN, VU). *Rhinobatos* *Rhinobatos* should not have a capital letter. You could potentially bold the species that have a high number of individuals caught within versus outside of MPAs.

Thanks for the useful suggestions. We modified the figure as requested by the referee. We created new barplots comparing the three IUCN categories here considered (Threatened, Low Risk and Data Deficient). Thanks also for spotting the capital letter in *Rhinobatos rhinobatos*, this has now been fixed.

Figure 3 – same suggestion. Make text bigger.

Revised as suggested.

L50 versus L50 choose one and make consistent through text

We thank the referee for the suggestion. We left L₅₀ throughout the MS.

Line 218 – I don't think you can say the restrictions on human use lead to a "increase" of density and biomass as you haven't showed that through time. Rather you can say it is correlated with. Were these MPAs placed there because they are important to sharks? Could fisheries be targeting sharks within MPAs and would that influence you CPUE estimates?

Thanks for the useful comment. We rephrased the sentence as suggested by the reviewer. Concerning the points raised by the referee, generally MPAs in the Mediterranean are established for wider goals such as "biodiversity protection", but in some cases can have specific goals related to the protection of some species, such as monk seal, turtles etc. According to the information available, there are no MPAs specifically established for elasmobranchs in the Mediterranean and Black Seas and the lack of legislation on this issue would come mainly from a lack of knowledge on critical habitats of this group (Bradai et al. 2012). None of the MPAs considered has been established specifically for shark conservation or has shark conservation among the conservation goals. We are actually confident that this is applicable to all other MPAs currently present in the Mediterranean Sea. Also, as reported in the introduction, elasmobranchs are not target species for SSF in the Mediterranean Sea and fishers do not use specific gears to catch elasmobranchs, and we have no elements suggesting fishing operations specifically targeting sharks within the MPAs.

Bradai, M.N.; Saidi, B.; Enajjar, S. Elasmobranchs of the Mediterranean and Black Sea: Status, Ecology and Biology; Bibliographic analysis; Studies and Reviews-General Fisheries Commission for the Mediterranean; No. 91; FAO: Rome, Italy, 2012.

I suggest being cautious about saying 'abundance' in place of CPUE (biomass or abundance). Although a very useful way to infer abundance the catch statistics are not fisheries independent estimates of 'true' abundance.

We thank the referee for this useful comment, with which we totally agree. We added a cautionary statement that reads: "although our estimates are fishery dependent, and assuming absence of hyperstability or hyperdepletion, CPUE could be considered a proxy of abundance and biomass at sea" (line 252-253 in the clean version). We also ensured to use the term "NCPUE", rather than "density", all the times we refer to catch per unit of effort in terms of density.

Discussion

You introduce fully protected MPAs and PPAs in Figure 1 but don't discuss it again. What I think is interesting is that the fully protected MPAs have LC species but no VU, EN, or CR species. These MPAs are the ones that would lead to an increase of abundance and biomass whereas the PPAs are the ones with high threatened catches. Implementing regulations in the PPAs with high shark catches could be a conservation win.

Thanks for the comment. We apologize for the lack of clarity. We introduced the term FPAs only to specify that two (Cote Bleue and Cap Roux) out of the 11 MPAs here considered included only Fully Protected Areas. As answered above to another question, we did not monitor any commercial catch inside the FPAs as fishing is always forbidden in these areas. For the two fully protected MPAs (Cote Bleue and Cap Roux), we only monitored catches from unprotected areas around the MPAs. Concerning the last part of the comment of the referee, we totally agree, in fact, in the last chapter of the discussion, we stressed how shark-oriented regulations of SSF within the PPAs are needed to ensure MPAs better meet their conservation objectives.

I think a paragraph discussing how the catch statistics were collected and how that could influence your results would be useful. Are fishers targeting sharks in PPAs versus UPAs? Are these PPAs established to protect sharks?

We thank the reviewer for this comment. As also commented above, elasmobranch species are not primary target species for SSF in the Mediterranean Sea and, for this reason, fishers do not use specific gears to catch elasmobranchs. This applies both to PPAs and UPAs, therefore we do not expect any influence of different fishing tactics/behaviour in affecting comparison between different protection levels. In this sense, we point out that the catches monitored in our study are commercial SSF catches, that were randomly sampled (i.e., interspersed) over the period of investigation. This is now reported in the revised version of the MS. Concerning the last sentence, the PPAs (and in general the MPAs) in the Mediterranean Sea, are area-based tools for marine biodiversity conservation broadly, and none of them has been established for the specific protection of elasmobranch species. As requested by the referee we included this point in the introduction.

Reviewer #3 (Remarks to the Author):

General comments

The study describes the assessment of elasmobranch catch in small-scale fisheries in the Mediterranean, comparing catch rates in areas of differing protection status. The study consists of a comprehensive dataset spanning 11 locations within six countries. I think the premise of the paper is sound and on a subject matter in need of attention, especially in an area such as the Mediterranean. I do feel however, the description and presentation of methods and results are lacking and do not provide enough evidence to substantiate the main conclusions. I think the lack of detail on the fisheries themselves is a major issue when comparing catch rates. There is little information on whether the fishers differ across countries, the gears being deployed, and key elements of effort such as soak time, and number of sets – which will hugely influence standardising and comparing catch rates.

We thank the referee for the thorough review and the effort she/he put in providing us very useful suggestions and comments. Many of her/his comments have really helped us to improve the manuscript. As requested, below we detailed some of the information suggested by the referee.

- I don't feel the methods are fully described, especially with regards data collection or assessment of the fishery itself.

We thank for the useful comment. We added an extensive description of the methodology adopted to collect the data (see also response to reviewer 1's comments). For a sake of simplicity, we created a new table including the restrictions and features of fishery operations; please, see Table S6 in Supplementary materials.

- I'm not sure the data presented necessarily back up conclusions and statements made.

We believe that this comment had positive intentions, in order to let us improving the manuscript, but we find very difficult to use this comment in a constructive way, being quite general. There is no reference of which specific data the reviewer is referring to or which conclusions and statements are not appropriate. In fact, we do not think that the comment is referred to the entire set of results and discussion provided, especially in the light of the appreciations made by the other 2 reviewers. In this perspective we carefully revised the entire MS making sure that all the conclusions are backed up by results.

- I don't feel the study is novel enough for Nature Communications.

We thank the referee for this comment that allow us to highlight the strength and the novelty of our study. Nowadays, at our knowledge, this is the only study comparing SSF catches of elasmobranchs between MPAs and UPAs. In addition, one of the strengths of this work is, surely, related to the scale at which we carried out the study. These aspects have been highlighted by the other two referees; particularly, the referee 1# wrote: "Data on small-scale fisheries are generally scarce, particularly at the scale seen in this study. That makes this an important study for shark and ray conservation on the Mediterranean Sea, which is itself a site of priority given the former abundance and species diversity in the area, but the high human impact over recent decades. The study appears to have been well carried out, the methods are robust and the conclusions supported by the results". Referee 2#: "The authors have presented excellent and novel research illuminating catch composition and distribution of elasmobranchs in a hotspot of elasmobranch threat".

- The absence of some key features of catch effort are very important to analysis.

We thank the referee for this comment. Actually information about net length (one key element in assessing fishing effort in SSFs) has been collected and included in our analyses, either as offset or to calculate catch per unit of effort. For what concerns other elements determining fishing effort, such as net soaking time and mesh size, we assumed they were randomly interspersed between the 2 conditions (protected vs unprotected) based on the fact that the same fishers were operating both in PPAs and UPAs at each location, using the same gears. However, based on the comment of the referee we tested the potential variability of these features between protection levels and we did not find any statistical differences between PPA and UPA protection levels in net soaking time ($X^2 = 3.1203$, $Df = 1$, $p_{\text{value}} = 0.08$) nor mesh size ($X^2 = 1.8953$, $Df = 1$, $p_{\text{value}} = 0.1686$). For these reasons these variables were not formally included in our analyses. For the sake of clarity and to fully report our data we also created a new table with the features of SSF operations at each sampling locations (please, see the Table S6 in Supplementary materials).

- Presentation and explanation of figures is limited.

Thanks for this comment. We have modified figures 1, 2, 3 to improve readability, following the comments provided by all 3 reviewers. For each figure, we improved the caption, adding relevant information to the comprehension of the figure. We also prepared a new figure (Fig. S1) about the number of fishing operations monitored in each location, that can be found in Supplementary Materials. In addition, we reworked the manuscript to improve the explanation of each figure within the main text.

- Little information on what constitutes a PPA – what are the restrictions? Do they differ across countries?

We thank the reviewer for this very pertinent comment that was raised also by another reviewer. PPAs (partially protected areas), as recently reported in Grorud-Colvert et al, 2021 (published in *Science*), include a range of spatially-explicit conservation tools characterised by different protection levels ranging from ‘minimal to high’ protection, established for conservation purposes, where some human uses (including extractive ones) are allowed but more or less strictly regulated, compared to FPAs where all extractive uses are forbidden or surrounding unprotected areas where fishing is exerted according to national laws. In particular, in our study, PPAs are in all the cases portions of multiple-use MPAs where SSF is allowed but strictly regulated. Specifically, fishing regulations in the PPAs considered include a variety of measures: limited entry, gear restrictions (e.g. mesh size or net length regulations), time restrictions (e.g. the use of certain gears only in specific seasons), territorial use rights for fishers and/or fish size limits. This information was included in the new version of the MS, and were summarized in a new table (Table S6, supplementary material) reporting the restrictions in each PPA.

- MPAs are mentioned but not compared. I understand there are fewer, but may have been interesting to see if there is a scale from no protection to full protection in terms of catch.

Thank you for this comment. As specified in previous responses to other referees’ comments, we highlighted that we monitored commercial small-scale fishery catches. Thus, given that fishing activities are not allowed within the FPAs, no commercial catches occurred and were monitored within those zones. We assessed commercial catches in 11 Mediterranean locations, all including a MPA. In 9 out of 11 locations, we monitored commercial catches from inside PPAs and outside. In the remaining two locations (Cote Bleue and Cap Roux) the MPAs include only FPAs (where no fishing is allowed), thus for these 2 MPAs, data are referred to fishing operations occurred in the UPAs outside the MPA. To make this even clearer, we added a new figure (Fig S1) showing the number of fishing operations we monitored in PPA and UPA at each location. In addition, we reworked the text of the MS to enhance clarity about this point.

- The abstract headlines are that more elasmobranchs were caught in PPAs compared to UPAs, and that this means protection can have benefits but SSF are risk to conservation. I see the point the authors are making, but I think that the fact that <1 shark per operation is being caught, which is very low, suggests intensive fishing likely already depleted many populations. I would also argue that the main finding lies in the higher number of immature individuals being caught in PPAs – have these PPAs been placed in nursery grounds?

We thank the reviewer for this very pertinent comment. The fact that elasmobranch catches are very reduced is probably the result of multiple pressures affecting Mediterranean coastal ecosystems, with historical overfishing probably playing a major role as already highlighted by other studies (Dulvy et al. 2014; Cashion et al. 2019). This is now discussed in the revised version of the MS. The PPAs, and more in general the MPAs, were not established to specifically protect elasmobranchs, and therefore we should exclude that PPAs have been purposively placed to protect nursery grounds. If fishing operations have been carried out also in nursery grounds, this would be independent of protection level. In this perspective, our sampling was carried in coastal areas at comparable depth ranges and habitats in all the locations and therefore we should assume the same probability for fishing operations to be carried out in nursery grounds both within and outside MPAs. Therefore, we can speculate that the higher proportion of immature individuals caught in PPAs is probably related to the effect of protection on these species.

Dulvy, N. K., Fowler, S. L., Musick, J. A., Cavanagh, R. D., Kyne, P. M., Harrison, L. R., Carlson, J. K., Davidson, L. N. K., Fordham, S. V., Francis, M. P., Pollock, C. M., Simpfendorfer, C. A., Burgess, G. H., Carpenter, K. E., Compagno, L. J. V., Ebert, D. A., Gibson, C., Heupel, M. R., Livingstone, S. R., Sanciangco, J. C., Stevens, J. D., Valenti, S. & White, W. T. (2014). *Extinction risk and conservation of the world's sharks and rays*. *eLife* 3, e00590. doi: 10.7554/eLife.00590

Cashion, M. S., Baily, N., & Pauly, D. (2019). *Official catch data underrepresent shark and ray taxa caught in Mediterranean and Black Sea fisheries. Marine Policy, 105, 1–9.*
<https://doi.org/10.1016/j.marpol.2019.02.041>

Specific comments

Title – Endangered should be changed to threatened as the descriptions used grouped IUCN classifications. Thank you for the suggestion. We modified the title as suggested also by referee 1# and 2#.

Line 15 – “Highly threatened” is used, what is this classification, it is not used elsewhere. Is this the same as the IUCN classification for threatened, and what the authors declare “high risk” later in the manuscript? Thank you for the comment. As suggested also by the other two referees we revised the text to be sure to align with IUCN terminology (“threatened” and its acronym “TH”) throughout the entire MS.

Lines 27-29 - This is not the only reason. SSF, which are under reported and are a source of huge shark catch.

Thank you for the useful comment. However, we refer to “Illegal, Unreported and Unregulated (IUU) catches” with this category including catches from all fishery segments, including also SSF. In the abstract we prefer to keep this general, in order to succinctly refer to all the relevant fisheries.

Line 40 - Not just offshore. Reported catch of elasmobranchs largely originates from industrial fleets, however much of this occurs in coastal waters on the continental shelf.

Thanks for your comment. We deleted “offshore water” from the text.

Lines 57-58 - Global estimates are between 5-7.5% of the ocean being in protected areas, below 10% target and way off new target of 30%. How much is 29% of 5%, and how much area. Should note that there are no shark sanctuaries in the Atlantic outside the Caribbean Sea.

We thank the reviewer for the comment, we further investigated this aspect and revised the text. Following the data from mpatlas (www.mpatlas.org), the 2.8% of global ocean is fully protected, 29% of that is dedicated to shark (Davidson and Dulvy 2017), with therefore an overall 0.81% of the global ocean (corresponding to 2,900,000 km²) dedicated to shark protection. We now clarified this in the revised version of the MS (lines 57-60).

Line 88 – Need more detail on the gear. What type of fixed net, mesh size, how is it fixed i.e. demersal set or set from the surface. What is the target species for this fishery?

Thank you for the comment. All the gears used by fishers during the fishing operations included in our study were fixed nets, typical of Mediterranean SSFs. Nets were predominantly trammel nets (in about 95% of the fishing operations monitored) with a minority of gillnets (4%) and combined trammel-gill nets (1%), these percentages well reflecting the use of these gears at Mediterranean level. These different types can be used to target different groups of fishes, but they are deployed and work similarly. They are anchored (and touch) to the bottom through a lead line, and are kept in a vertical position by a float line. Normally the height of the net is 3-4 m for trammel nets, 6-8 for gillnets and combined nets. Given the unbalanced proportion of the net type, we did not include this information in the analysis as practically meaningless for the model. Concerning mesh size, this depends on the type of net and the national/regional and local (i.e. within the PPAs) regulations. Concerning the target species, although some differences among locations exist, fixed nets are classical gears of multi-specific Mediterranean SSF and are used to target a variety of coastal species including scorpion fishes, mullids, sparids, labrids, pelagic fishes, flatfish, coastal cephalopods and lobsters. Further details on the gears used were added in the methods in the new version of the MS (lines 335-360 of the clean version) and a new table was created to summarize those, please see Table S6 in the Supplementary Materials. As specified in a previous reply we also tested for potential variability between protection levels (PPAs vs UPAs) in soaking time and mesh size, with this information now reported in the methods and in the supplementary materials of the revised version of the MS.

Line 96 – I would caution against claiming SSF to be the threat to elasmobranchs in the Mediterranean. These fisheries are catching quite low levels relatively based on the results and so SSF may exacerbate the situation or catch more immature individuals, but these comparisons are not presented here.

We thank the referee for this comment. Our statement wanted to highlight that SSF has the potential to be one of the threats to elasmobranchs in the Mediterranean, without implying SSF being the most relevant. We believe our results suggest that, by removing individuals (including immature ones), SSF has the potential to impact local populations of elasmobranchs. We rephrased the statement in order to make it more cautious and reflect our findings. Please, see lines 96-98 of the clean version.

Figure 1 – Are these locations where the catches were landed or fished (or both) as these are describing PPAs, and MPAs but not UPAs. Where is the composition of UPA catch?

- The caption states “i.e. Biomass CPUE”, this is confusing, are the data from BCPUE? As these may differ from composition by individuals.

Thank you for the comment. We apologize for the lack of clarity in the previous version of the MS. The figure 1 shows the percentage of IUCN categories of elasmobranchs caught in the 11 Mediterranean locations (each including an MPA and/or surrounding unprotected area) where SSF catches were sampled. In the new version of the MS, we added a figure showing the number of fishing operations we monitored at each location both in PPAs and UPAs (please see Fig. S1). We also changed “Biomass CPUE” with “BCPUE” as requested.

Line 105 – 1256 operations carried out – What constitutes an operation? How many sets were deployed per operation? Across what timeframe were these data collected? Soak time? Number of hours/days? Does time of day fishing change? Is <1 shark per operation a lot?

Thank you very much for the comment. We apologize for the lack of clarity in the previous version of the MS. We reworked the text in order to provide all the requested information and ensure clarity.

The term ‘Operation’ refers to a single net deployed in the water. Multiple operations could have been conducted by a single fisher during the same day. In most of the cases each fisher carried out 1 or 2 fishing operations per day, but in few cases some fishers carried out 5-6 operations per day. Time of the day for each operation could vary a lot depending on fishers’ habits, generally depending on local market habits. Following also the comment of another reviewer, we clarified which was the sampling timeframe. Specifically, SSF catches were sampled over 16 months, between June 2017 and October 2018, with catch dates assigned to exact boreal seasons.

As also anticipated in replies to previous comments, for the sake of clarity and to fully report our data, we added a new table to summarize the features of fishing operations at each location (see Table S6 in the Supplementary Materials).

About the last question (“Is <1 shark per operation a lot?”), we believe this is a tricky point, because in absolute number this could look as a low catch rate, but considering the number of fishing operations carried by the multitude of small-scale fishers in the Mediterranean Sea (more than 71.000 vessels operating in the Mediterranean Sea) overall this could sum-up to a large number of elasmobranchs fished. We believe that further studies should be carried out to expand our research and assess the overall catch and potential impact on local populations. This is discussed in the revised version of the MS.

Figure 2 – In my opinion this figure doesn’t work. I can see what the authors are trying to do to make an interesting plot, but I think a simpler presentation of the results would be more informative. Also the colours are not very distinguishable to colour-blindness.

Thanks for spotting this. We modified the figure as requested by referee 1# and 2#, who found the figure useful. The colours have been changed (we also changed the colours in the figure S2 to make them consistently) as requested by referee 3# and we added two more informative barplots as requested by referee 2#.

Line 134 – I’m not sure why the authors have redefined a category established by the IUCN. Species in VU, EN, and CR are classified collectively as “threatened”.

As replied in previous comments we have revised the text and used ‘threatened’ throughout the MS.

Lines 145-146 – Is this a little contradictory as many of the HR species are demersal too?

Thank you for the pertinent and stimulating comment. We reworked the sentence speculating that this pattern could be rather related to different home ranges. Specifically, rays (the most species of the low risk group) generally show a sedentary behaviour with limited home ranges and do not show habitat partitioning between juveniles and adults, differently from shark species (the most individuals of the threatened group) in which adults generally show a philopatric behaviour (for example *Mustelus* spp.) moving away from coastal

areas where generally SSF do not operate. For this reason, likely mature and immature individuals from the Low Risk group have been fished in similar proportion by SSF operating in coastal areas, while this did not happen for species of the threatened group. This has now been discussed in the revised version of the MS (see lines 167-175).

Figure 3 – This figure needs more explanation. I think the two smaller graphs should be bigger and aligned to the right of the main panel, so that they are half the height of the main figure.

Thanks for the useful suggestion. We fixed the figure as requested by the referee.

Lines 185-186 – This statement only applies to HR and DD species – I think this should be made more explicit.

Done. Thanks for the suggestion.

Line 192 – I think using a word like confirmed is a little strong when presenting these results.

Thanks for spotting this. As we removed the figure and the previous sentence, we replaced ‘confirmed’ with ‘suggested’.

Lines 192-193 – Why are test results not presented here?

We decided to show the tables in Supplementary materials to make the main text easily readable, however, in the revised version of the MS we added the results of protection in brackets.

Lines 195-197 – I assume the relationships described are non-significant? Can the test statistics be presented here?

Thanks for the comment. All relationships with the variables the referee is referring here (chl_a, SST, human pressure) were significant and they are reported in Table S5. We revised the text to clarify this part and added the results in brackets.

Line 214 – Is density the right description? Is it occurrence or presence maybe?

We thank the referee for this comment. Based also on a suggestion by referee 2 we replace density and biomass with NCPUE and BCPUE to make this consistent throughout the text.

Lines 241-251 – I think this paragraph is too broad and too vague, I think it needs to be incorporated with the next section where examples are given for strategies rather than simply stating conservation needs to be done.

We agree with this comment and we incorporated this paragraph with following section.

Line 243 – “Half of the elasmobranchs are Critically Endangered” should be “half the elasmobranch species are Critically Endangered”.

Revised according to referee’s suggestion. Thank you very much.

Line 275 – Consistency of terms – “highly threatened” has not been defined, either refer to IUCN classifications or the High Risk category defined and used throughout.

Thank you very much for spotting this. As specified above we removed “highly threatened” and used the correct term (“Threatened”) throughout.

Line 310 – Again details on the fishery and fishing operation are needed.

We thank the reviewer for this comment, and we apologize for the lack of details in the previous version of the MS. We added more details about fishery and fishing operations (lines 342-360 in the clean version), and added a new table to summarize those, please see Table S6 in the Supplementary Materials.

Line 312 – More information required for photo-sampling technique. Were individuals photographed separately, or more general pictures taken to count and identify later?

We apologize for the lack of clarity. We added all relevant information describing the photo-sampling methodology in the new version of the MS (lines 348-360 in the clean version). Please, for further details see also reply to referee 1’s comment.

Line 316 – What were the types and lengths of the nets? Also are there other details on the fishery gathered – soak time? Number of sets? Duration?

We thank the reviewer for this comment. As replied in a similar comment above, the information indicated by the reviewer were recorded for each fishing operation, and this has now reported in the revised version of the MS in a synthetic table (Table S6 in the Supplementary Materials).

Line 326-328 – Are these done nominally? To properly model a rate, an offset should be used to account for variation in a defined effort metric.

Thanks for the very pertinent comment. Based on this comment and on others provided also by the referee we run additional analyses in order to compare outputs of the models with catch standardized per unit of effort and with catch not standardized per unit of effort but including the offset, and finally select the model with the best fit based on AIC and residuals. As suggested by the referee, therefore, we performed an analysis using the error family distribution “Zero inflated Poisson” on count of individuals with the net length used as offset and we compared the AIC and the residuals with the model we used to analyse the data. The AIC results have shown a bit lower value for ZIP (2030) compared to ZAGA (2100) distribution; however, as AIC values do not provide information about model quality, the residuals check for independence and identical distribution, using the worm plot function were also assessed and they highlighted a high difference of the deviance (normality) of the ZIP distribution (please see the figure below); therefore, as the results between the two distributions were similar (please see the figure below) we decided to keep the distribution showing better residuals (ZAGA).

Residuals of the ZIP (panel on the left) and ZAGA (panel on the right) models

a) ZAGA distribution

B) ZIP distribution

The effect of different variables on number of individuals Caught per Unit Effort (NCPUE, kg /1000 net) of elasmobranchs, based on the GAMLSS model. a) Zaga Distribution, b) Zip Distribution

Line 338 – What resolution were environmental variables acquired at? Any standardising? Would daily rasters have given a better account of environmental conditions?

We apologize for the lack of details in the previous version of the MS. All environmental variables were acquired from the Copernicus database and extracted at the highest resolution possible (spatial: 0.042 degree; temporal: daily value) for the study period. This details have now been included in the revised version of the MS.

Line 340 – Why was bathymetric depth used and not depth of gear?

We thank the reviewer for this comment, and we apologize for the lack of clarity. We definitely used the approximate depth of fishing operations as reported by fishers (this is now clarified in the revised version of the MS). As specified in a previous reply, all nets have been deployed on the bottom and therefore bathymetric depth identifies also the depth at which the net was operating.

Line 340 – What is the layer of human pressure?

Human pressure was extrapolated from Micheli et 2013, we added this information in the text.

Lines 352-353 – I'm not sure what this model is referring to. The length data described is of proportions of mature and immature individuals. What is the zero-inflated count of?

Thanks for spotting this. As did also for comparing different models (with and without offset), we performed the GAMLSS analysis using error family distribution "Zero inflated Poisson" (ZIP) on count of immature and mature individuals with the net length used as an offset and compared with the results presented in the previous version of the manuscript (GAMLSS-ZAGA). For each analysis we used the AIC results and the graphic inspection of the residuals to know the best approach; for this purpose, we added the new results in the main text and a table showing all the AICs in Supplementary Materials (Table S9) and modified the text based on the new approach. The new results, are overall in line with the ones presented in the previous version of the manuscript. Particularly, immature individuals of Threatened and Data Deficient species were more frequently captured in PPA than UPA. Low Risk species did not show any difference for immature and mature.

Line 364 – I would again argue the best approach would be to use an offset to model the rate of CPUE. Type of net was said to have been asked when interviewing the fishers, is this not included as a variable in the models?

Thanks for this pertinent comment. As answered above, we compared ZIP (with an offset) and ZAGA models as suggested by the referee; Residuals and AIC results highlighted that ZAGA models of CPUE were the best approaches to analyze the data. As specified above, nets were predominantly trammel nets, in about 95% of the fishing operations monitored, and the remaining 4% were gillnets or 1% combined trammel-gill nets, these percentages well reflecting the use of these gears at Mediterranean level.

In conclusion, we did all our best to comply with the referees suggestions, and we hope the revised version could match the journal's standard and be suitable for publication, but we will be happy to consider other changes according to the indications you may wish to provide us.

REVIEWER COMMENTS

Reviewer #1 (Remarks to the Author):

Thank you for considering my suggestions and modifying your ms in response where you found my comments constructive and helpful. It looks to me like you have done a good job of responding to all of the reviewer comments. I hope to see your nice paper in print soon.

I just found a few minor issues as I was reading through the revision (line numbers refer to the clean version):

Line 149: I don't think you have defined 'TH' in the text before this point to mean threatened?

Line 361: There is an 'HR' here that should have been changed to 'TH'.

Reference 4 and 9 are the same.

Best wishes,

Richard B. Sherley

Reviewer #2 (Remarks to the Author):

Dear authors,

Thank you for your efforts and for your thoughtful responses to the suggested revisions.

I found your revisions clarified the text and strengthened the manuscript.

I thought your study met the objectives that you identified including characterizing the landings from SSF, investigating CPUE difference between PPA and UPAs, disentangling protection versus environmental covariates.

My final suggestions remain with clarifying some of the interpretations and the text around the role of PPAs and UPAs. Specifically, I think reporting the numbers caught in addition to CPUE would be useful. CPUE is used as an index of abundance however understanding the magnitude of the catches or effort would be useful to interpret conservation implications for both inside and outside of the PPAs. For example, on line 264 – the text reads “although elasmobranchs were more frequently caught in the UPAs” and on line 104: “Due to the probably high fishing pressure in UPAs than in PPAs we hypothesized that elasmobranch CPUE is higher in PPAs than in UPAs” which your results support. However, if effort and catches are higher outside PPAs that would be useful management information as opposed to lower catch but higher CPUE inside PPAs. Also, the relatively higher number of immature individuals captured with PPAs could have the counter narrative that more mature individuals are being captured outside of PPAs and that could represent a more important conservation priority. See below for more details.

Text such as Line 116: “Our study provides evidence on the role of PPAs in protecting coastal elasmobranch species and highlights that SSFs represent a threat for these species, also inside Mediterranean MPAs, suggesting the critical need for careful management measures.”

On line 99 of the edit version you state “no studies to date have focused on the potential role of PPAs to protect the elasmobranch species that were once widespread throughout the Mediterranean coastline”

From Figure 2. Is the decimal value in the circle plot the percentage of all of the PPA and UPA catches that is that species? I think I misunderstood this figure in my previous revision. So I think now what would be more useful is to have highlight the species that are caught more (by absolute numbers) inside versus outside PPAs. Or you could have two N values, the value for UPA and PPA. You explore the BCPUE and NCPUE in the second part of the figure but you could also explore the absolute numbers here. The idea being is catching fewer sharks the conservation goal or is reducing effort that will have the biggest reduction in catches the conservation goal?

Line 132 – “Immature threatened elasmobranchs are caught the most by SSFs” – I recommend changing this as this paragraph is more about the composition of catches across all SSF operations. Also ‘caught most’ is hard term to parse with respect to CPUE/BCPUE or absolute numbers.

I would suggest a more cautious or narrative for Figure 3 that considers that more mature individuals are being fished outside of PPAs (in the UPAs) and that could be the bigger conservation problem – see Prince et al on gauntlet fisheries where they state “Furthermore this management strategy proves to be most effective with the species considered to be least productive, those with greatest longevity.”. I think this perspective could be useful in your results/discussion especially here with respect to discussing a minimum, landing size. See line 240 and how this alternative perspective would influence your recommendations and interpretation.

Line 264: Your models show that elasmobranchs were more frequently caught in the UPAs but B/NCPUE is higher inside the PPAs – does this change some of the discussion? Biomass caught of species caught from your Figure 1 show many species are caught outside of PPAs and this makes me cautious about implying a weighting to the importance of SSFs versus other types of fishing that occurs inside or outside of PPAs.

One line 267 Finding a positive relationship between NCPUE and human pressure is not what I would have expected – is this supposed to be negative?

It could be useful to quickly describe what your measure of ‘human pressure’ was.

Line 348 Dipturus -> Dipturus

Line 350 – “the removal of threatened elasmobranchs from inside PPAs by SSF appears to reveal a conservation paradox with threatened species being more abundantly fished at an immature stage within PPAs.” I am not sure I agree with this statement given I think fishing the adults outside the PPAs to potentially be a bigger conservation issue. Also if the PPAs are working and biomass and numbers are increasing within PPAs, sustainable removal of fish would be the end goal for recovery. I therefore think your conclusions could be more focused on the balance of conservation issues both inside and outside PPAs. This is where the discussion of absolute catch numbers would also be interesting. If CPUE is low but absolute catch count is still relatively high outside of PPAs that is a different management opportunity than the high CPUE but low absolute catch count of sharks inside PPAs?

Line 381- capitalize IUCN cats.

Last paragraph starting at 394. I would also include sustainable management of the matrix outside of MPAs. MPAs are providing some benefit here however the high number of sharks caught outside of the MPAs seems to be a conservation priority as well.

Reviewer #3 (Remarks to the Author):

General comments

This is the second time seeing this paper, and I thank the authors for their efforts to improve the manuscript and address as many queries posed as possible. I do, however, still have some concerns over the analysis and presentation of the results. My main concern is the description and application of some of the analyses. The descriptions of what approaches were applied to different aspects of the data feel a little muddled, and in some instances, I am not sure if the techniques employed are the most appropriate. I think outlining specific aims/questions, then describing which data and which methods were used to answer those aims would help the reader follow the process including why the analytical approach was chosen and what structure does it take.

Specific comments

Lines 26-28 - Do you mean annually? The reference cited for this is from 2015, is there a more recent paper to back up catch rates from 2014 onwards?

Lines 58-60 - This sentence reads as though the designation of <1% of the ocean as protected for sharks is positive? Perhaps there needs a statement contextualising this as this is unlikely to achieve conservation goals for these species.

Line 79 – Under normal fishing conditions if encountered are sharks retained?

Line 128 - High relative to other studies? Or something else?

Lines 148-149 – Any formal test of whether more immature individuals were caught inside vs outside PAs?

Lines 150-151 – I'm not sure of the value of reclassifying IUCN categories into High Risk/Low Risk as the categories can be compared as they are and then are comparable to other studies.

Lines 173-176 – These results should be presented in the main manuscript.

Lines 182-186 – This isn't fully described in the methods and appears to test the nominal CPUE with the same variables as in the other models but isn't presented in a way that shows its added value to the results or discussed in detail in the manuscript.

Line 215 – Just states CPUE, is this as you are referring to both NCPUE and BCPUE?

Line 340-343 – Was it not possible to sex individuals from the photos to do species and sex specific size at maturity?

Lines 363-374 – From the description it looks like nominal NCPUE and BCPUE are being modelled as positive continuous variables and as such are these able to be considered as zero-inflated? Or are these approaches based on the zeros creating a skewed data set? This is also the analysis where an offset would best be used as this allows for the raw numbers of sharks being caught to be modelled as a rate per net length, and if zero-inflated, then ZIP Or ZINB models may be appropriate.

Lines 380-381 – What was the structure of the model for immature vs mature individuals? It is stated as using a ZAGA with an offset for net length. Was the offset the log of the net length? Are all the nets the same height? If not, perhaps m² is a better metric for effort? Also is a Gamma distribution the most suitable error structure for count data? The supplementary table for this analysis says the proportions of immature/mature individuals was modelled?

Line 386 – Was there a formal test for zero-inflation? Plotting numbers/proportions of zeros doesn't inform us whether these are more than would be expected.

Lines 414-416 – What is the definition of a species composition response that is being tested in the pRDA?

Figures – I think figures for the outputs from the analysis of what is driving CPUE should be in the main manuscript.

Table S9 – There isn't an explanation that two model types were compared for this in the main text and so isn't clear what this table is showing. The description of the structure of the ZIP or ZAGA is not clear. If using proportion of immature versus mature, I'm not sure either of these error structures are suitable. The details of the model would be good here, similar to Table S10 to show what was being tested and structure of formula.

Table S10 - It would read better if the models were presented from lowest AIC to largest, as these are how they are ranked. Why are some models in bold, it is not explained in the caption?

Please find attached our revised version of the manuscript by Di Lorenzo, Calò, Di Franco et al. entitled: "Small-scale fisheries catch more threatened elasmobranchs inside partially protected areas than in unprotected areas", for consideration for publication in Nature Communications.

We thank again the referees for their time, feedback and very useful comments, which have helped us to significantly improve our MS.

We addressed the great majority of the referees' comments and requests for clarification. In the very few cases where we preferred not to modify the MS as suggested by the referees, we have provided justifications for our decisions. Point-by-point responses are provided below in blue.

Also in this second round of revisions we include two versions of the main text and supplementary materials: one with track changes for you to check our edits, and a second clean file.

I look forward to your response.

Reviewer #1

Thank you for considering my suggestions and modifying your ms in response where you found my comments constructive and helpful. It looks to me like you have done a good job of responding to all of the reviewer comments. I hope to see your nice paper in print soon.

We thank the reviewer again for appreciating our work.

I just found a few minor issues as I was reading through the revision (line numbers refer to the clean version):

Line 149: I don't think you have defined 'TH' in the text before this point to mean threatened?

Thanks for this comment. In the previous version of the ms we defined TH at lines 120. Now we moved this at line 109 (clean file), and capitalized the term (fixing this from the previous version).

Line 361: There is an 'HR' here that should have been changed to 'TH'.

Thanks for spotting this. Done as suggested.

Reference 4 and 9 are the same.

Thanks for spotting this. We now fixed this by modifying reference 9.

Reviewer #2

Dear authors,

Thank you for your efforts and for your thoughtful responses to the suggested revisions. I found your revisions clarified the text and strengthened the manuscript. I thought your study met the objectives that you identified including characterizing the landings from SSF, investigating CPUE difference between PPA and UPAs, disentangling protection versus environmental covariates.

We thank the reviewer for appreciating our work.

My final suggestions remain with clarifying some of the interpretations and the text around the role of PPAs and UPAs. Specifically, I think reporting the numbers caught in addition to CPUE would be useful. CPUE is used as an index of abundance however understanding the magnitude of the catches or effort would be useful to interpret conservation implications for both inside and outside of the PPAs. For example, on line 264 – the text reads "although elasmobranchs were more frequently caught in the UPAs" and on line 104: "Due to the probably high fishing pressure in UPAs than in PPAs we hypothesized that elasmobranch CPUE is higher in PPAs than in UPAs" which your results support. However, if effort and catches are higher

outside PPAs that would be useful management information as opposed to lower catch but higher CPUE inside PPAs.

We thank the referee for this very pertinent and useful comment that made us realize that some sentences in the previous version of the ms may have not been clear enough. We apologize for this and we have now reworked the text to make it clearer. Particularly, in our analyses we used family distribution “Zero Adjusted Gamma (ZAGA)” that allows to model response variables with excess of zeros. Our two-part model analysis consists of a binomial (*logit*, presence/absence) model which predicts the probability of fishing operations with at least one elasmobranch individual in the catch, and a positive (truncated) abundance (*log*, where all zeroes are excluded), model which predicts the potential density or biomass of elasmobranchs (Stasinopoulos D. M. & Rigby R.A. 2007). Therefore, our results showed that the probability of elasmobranch catches (i.e. the number of fishing operations in which at least one individual was fished) using B/NCPUE data were higher in UPAs than in PPAs, while the gamma part, concerning the continuous values of B/NCPUE, showed the highest values of BCPUE and NCPUE were recorded in PPAs. Please, see line 177-184 (clean file); and see line 402-407 (clean file) for methods descriptions.

Concerning the number of individuals fished, we added a sentence to describe the number of elasmobranchs caught in PPAs and UPAs. Please, see line 170-173 (clean file).

It is however important to highlight that, unfortunately, we cannot get a yield estimation as fishing effort data of SSF are not available in the Mediterranean, as in the vast majority of cases globally (due to the absence of tracking devices on small fishing vessels). This is currently a major gap, and surely, this is a good point of discussion because such information could be crucial to improve elasmobranch conservation and fisheries management in general. We added a sentence in the introduction (line 86-89, clean file) and conclusion (line 270-272, clean file) to emphasize the importance to collect fishing effort data.

Also, the relatively higher number of immature individuals captured with PPAs could have the counter narrative that more mature individuals are being captured outside of PPAs and that could represent a more important conservation priority. See below for more details. Text such as Line 116: “Our study provides evidence on the role of PPAs in protecting coastal elasmobranch species and highlights that SSFs represent a threat for these species, also inside Mediterranean MPAs, suggesting the critical need for careful management measures.” On line 99 of the edit version you state “no studies to date have focused on the potential role of PPAs to protect the elasmobranch species that were once widespread throughout the Mediterranean coastline”

We thank the reviewer for this comment. However, our results suggest that a higher number of immature individuals caught, or a higher catch probability, within PPAs does not necessarily imply that a higher number of mature individuals are caught in UPAs (or that they are more frequently caught). This can be observed in Figure 3, and in particular 3c. Mature individuals are less frequent than immature ones both in the PPAs and in the UPAs. This is further confirmed by Figure S5 in the supplementary materials: on average, the number of mature individuals is never higher in the UPAs compared to PPAs (regardless the IUCN category). From this perspective, we believe that this interesting insight from the reviewer cannot be applied to the results of our study. However, we recognize the importance of that as a conservation issue and we decided to discuss it in the conclusion of the new version of the ms (line 253-261, clean file). Additionally, we clarified in the ms that we performed separate analyses for mature caught in PPAs and UPAs and immature caught in PPAs and UPAs (line 414-420, clean file).

From Figure 2. Is the decimal value in the circle plot the percentage of all of the PPA and UPA catches that is that species? I think I misunderstood this figure in my previous revision. So I think now what would be more useful is to have highlight the species that are caught more (by absolute numbers) inside versus outside PPAs. Or you could have two N values, the value for UPA and PPA. You explore the BCPUE and NCPUE in the second part of the figure but you could also explore the absolute numbers here. The idea

being is catching fewer sharks the conservation goal or is reducing effort that will have the biggest reduction in catches the conservation goal?

Thanks for your comment. The decimal value in the circle plot represents the percentage of the fishing operations in which at least one individual was fished, in order to align this with the information provided by the analyses (please see previous comment about ZAGA). We also report NCPUE and BCPUE as these are also aligned with the output of the models, and these value are useful to be plotted since they are standardized (i.e. g. and n. /1000 m of net) and could be compared with data from other studies. We agree with the referee to highlight the absolute numbers of elasmobranch catches and we added those in the section “Higher elasmobranch catches per unit of effort inside partially protected areas” (please, see line 170-177, clean file). About the last point, as discussed in a previous reply, unfortunately we cannot get a yield estimation as fishing effort data of SSF are not available in the Mediterranean, as in the vast majority of cases globally. This is a major gap currently, and surely, this is a good point of discussion because such information could be crucial to improve the elasmobranch conservation. We added a sentence in the conclusion to emphasize the importance to collect fishing effort data. Please, see line 274-277 (clean file).

Line 132 – “Immature threatened elasmobranchs are caught the most by SSFs” – I recommend changing this as this paragraph is more about the composition of catches across all SSF operations. Also ‘caught most’ is hard term to parse with respect to CPUE/BCPUE or absolute numbers.

Thanks for this comment. We changed the title to make it more in line with our findings, and to express the concept that threatened species were the most abundant among the elasmobranch fished by SSF. This is highlighted by our results with NCPUE and BCPUE representing relative abundance of group of species, and showing the highest values for Threatened elasmobranchs. The title now reads “Elasmobranch catches by SSF are mostly represented by Threatened species”.

I would suggest a more cautious or narrative for Figure 3 that considers that more mature individuals are being fished outside of PPAs (in the UPAs) and that could be the bigger conservation problem – see Prince et al on gauntlet fisheries where they state “Furthermore this management strategy proves to be most effective with the species considered to be least productive, those with greatest longevity.”. I think this perspective could be useful in your results/discussion especially here with respect to discussing a minimum, landing size. See line 240 and how this alternative perspective would influence your recommendations and interpretation.

We thank the referee for this comment. As stated above in a previous response, our results do not indicate that more mature individuals are being fished in the UPAs. Actually, Figure 3c (in the main text) and S5 (in the supplementary materials) can help clarifying this aspect. In fact, it can be noticed that in the UPAs, the proportion of mature individuals is not higher than the immature ones. We, however, think that this management strategy is worth to note and we now discussed that at lines 255-261 (clean file) citing the paper by Prince (2005).

Line 264: Your models show that elasmobranchs were more frequently caught in the UPAs but B/NCPUE is higher inside the PPAs – does this change some of the discussion? Biomass caught of species caught from your Figure 1 show many species are caught outside of PPAs and this makes me cautious about implying a weighting to the importance of SSFs versus other types of fishing that occurs inside or outside of PPAs.

Thanks for this comment. As detailed above we changed the sentence as it was confusing. The sentence “more frequently caught in UPAs” refers to the number of SSF operations in which at least one elasmobranch individual was fished. The sentence now reads “Although the probability of having elasmobranchs in SSF catches (i.e. the proportion of fishing operations in which at least one individual was

present) was higher in UPAs than in PPAs, the BCPUE and the NCPUE were higher inside the PPAs". It is worth noting that all the analyses were performed using B/NCPUE.

Concerning the comment about the Figure 2, it shows the percentage of fishing operations where elasmobranchs were found at least once, according with the probability of occurrence as described above. We decided to keep this information as it could be easily replicated everywhere. The other types of fishing occurring in PPAs and UPAs, within similar depth ranges, are mainly recreational and illegal fishing. Elasmobranchs are not target species for either of these two fishing categories as they focus on high value commercial species (e.g. groupers, sea breams, amberjacks etc.). For these reasons we assume that SSF can play a greater role in fishing elasmobranchs.

One line 267 Finding a positive relationship between NCPUE and human pressure is not what I would have expected – is this supposed to be negative? It could be useful to quickly describe what your measure of 'human pressure' was.

Thanks for the comment. The measure of human impact we used is the one developed by Micheli et al. 2013 (<https://journals.plos.org/plosone/article?id=10.1371/journal.pone.0079889>). In the previous version of the ms this was reported in supplementary material. In the revised version, for the sake of clarity, we included the reference also in the main text and briefly explained what this measure refers to, as requested by the reviewer (see Lines 380-383, clean file). This measure of cumulative pressure is a compounded score that includes information related to a number of human drivers (e.g. fishery, pollution, population density, climate change). We found a positive relationship between the overall human impacts index considered and NCPUE, while no relationship between the index and BCPUE was detected. This evidence may suggest that the positive relationship detected for NCPUE is mainly determined by the presence of juveniles and/or small-sized species (that contribute in high numbers of individuals, but less in terms of biomass) in areas with high human impacts. In the absence of additional elements, we hypothesize that the counterintuitive relationship between NCPUE and human impacts may be related to a combination of factors including the removal of top predators in highly impacted areas, potentially releasing meso-predators (as many of the species observed in this study) and/or juveniles. This mechanism has been previously suggested (Ferretti et al. 2008), but opposite evidences have been also reported (Ferretti et al. 2013). These studies are however referred to species exploited by large scale fisheries, and to the best of our knowledge no previous evidences are available for small scale fisheries in coastal areas. This is now reported in the revised version of the manuscript.

Line 348 Diptururs -> Dipturus

Thanks for spotting this, we fixed that.

Line 350 – “the removal of threatened elasmobranchs from inside PPAs by SSF appears to reveal a conservation paradox with threatened species being more abundantly fished at an immature stage within PPAs.” I am not sure I agree with this statement given I think fishing the adults outside the PPAs to potentially be a bigger conservation issue. Also if the PPAs are working and biomass and numbers are increasing within PPAs, sustainable removal of fish would be the end goal for recovery. I therefore think your conclusions could be more focused on the balance of conservation issues both inside and outside PPAs. This is where the discussion of absolute catch numbers would also be interesting. If CPUE is low but absolute catch count is still relatively high outside of PPAs that is a different management opportunity than the high CPUE but low absolute catch count of sharks inside PPAs?

We thank the reviewer for this comment. As described in a previous reply, our results do not indicate that more mature individuals are being fished in the UPAs. However, we think that the referee comment is relevant and now we discussed this important management strategy at lines 265-261 (clean file).

Line 381- capitalize IUCN cats.

Thanks for the comment. IUCN cats were capitalized.

Last paragraph starting at 394. I would also include sustainable management of the matrix outside of MPAs. MPAs are providing some benefit here however the high number of sharks caught outside of the MPAs seems to be a conservation priority as well.

We agree with the reviewer and we have discussed that in the conclusion, please see lines 255-261 (clean file).

Reviewer #3 (Remarks to the Author):

General comments

This is the second time seeing this paper, and I thank the authors for their efforts to improve the manuscript and address as many queries posed as possible. I do, however, still have some concerns over the analysis and presentation of the results. My main concern is the description and application of some of the analyses. The descriptions of what approaches were applied to different aspects of the data feel a little muddled, and in some instances, I am not sure if the techniques employed are the most appropriate. I think outlining specific aims/questions, then describing which data and which methods were used to answer those aims would help the reader follow the process including why the analytical approach was chosen and what structure does it take.

We thank the referee for his comments. We checked and, when opportune, added details to the statistical approaches. As described below we are confident the techniques and approaches we used were appropriated for our data and aims. For the sake of simplicity, we decided to add a flow-chart showing the systematic approach applied for the selection of the best statistical model. You can find the new figure in the supplementary material (Fig. 6S) and we also referred to that in the main text. Please, see line 394-395 (clean file)

Specific comments

Lines 26-28 - Do you mean annually? The reference cited for this is from 2015, is there a more recent paper to back up catch rates from 2014 onwards?

Thanks for the comment. The referee is right and these are annual estimates. However, we revised the text to make it clearer. Now, we refer to the global elasmobranch catches peak occurred in 2003 before declining due to the overfishing.

Lines 58-60 - This sentence reads as though the designation of <1% of the ocean as protected for sharks is positive? Perhaps there needs a statement contextualising this as this is unlikely to achieve conservation goals for these species.

Thanks for the comment. We apologize for the lack of clarity, and we actually wanted to express that this coverage is probably far below the optimal. In the revised version of the ms we added a sentence to highlight that the current percentage of protected ocean is not enough to obtain suitable elasmobranch conservation goals as highlighted by Davidson and Dulvy 2017.

Line 79 – Under normal fishing conditions if encountered are sharks retained?

Yes, different shark species are usually retained when fished. Mediterranean SSF are multi-specific fisheries in which practically every accessory species caught is normally retained and landed, even when they contribute in a minimal way to fishers' revenues. Thus, also sharks that normally have a low economic value per kg, are retained and sold. We added this details in the MS (lines 78-79 in the clean file).

Line 128 - High relative to other studies? Or something else?

Thanks for the comment. We apologize for the lack of clarity. We were referring to the high number of species out of the total present in the Mediterranean Sea. The sentence now reads “Elasmobranch captures represented relatively a small percentage of the total catches in our study (both in terms of NCPUE and BCPUE); however, given the high number of elasmobranch species captured, on the total living in the Mediterranean Sea, and the considerable proportion of TH species fished, systematic assessment of SSFs catches should be implemented”.

Lines 148-149 – Any formal test of whether more immature individuals were caught inside vs outside PAs?

Thanks for this comment. As described in the manuscript, we performed GAMLSS models to test that. Please, find the results at lines 205-211 (clean file) and the description of the statistical approach at lines 414-419 in the clean file.

Lines 150-151 – I’m not sure of the value of reclassifying IUCN categories into High Risk/Low Risk as the categories can be compared as they are and then are comparable to other studies.

Thanks for the comment. The classification we used is in line with the one used in an IUCN red list categories edition (2012) and in some recent papers (Ripple et al. 2017, Atwood et al. 2020, Munstermann et al. 2021,). In the new version of the ms we replaced the terms ‘high risk’ and ‘low risk’ with ‘threatened’ and ‘nonthreatened’, respectively, to make the classification fully aligned to the one already present and accepted in the scientific literature. This is now specified in the revised version of the manuscript. Please, see lines 390-393 in the clean file.

Atwood, T. B., Valentine, S. A., Hammill, E., McCauley, D. J., Madin, E. M., Beard, K. H., & Pearse, W. D. (2020). Herbivores at the highest risk of extinction among mammals, birds, and reptiles. Science Advances, 6, eabb8458.

IUCN. (2012). IUCN Red List categories and criteria: Version 3.1. 2nd edition. Gland: IUCN.

Munstermann, M. J., Heim, N. A., McCauley, D. J., Payne, J. L., Upham, N. S., Wang, S. C., & Knope, M. L. (2021). A global ecological signal of extinction risk in terrestrial vertebrates. Conservation Biology.

Ripple, W. J., Wolf, C., Newsome, T. M., Hoffmann, M., Wirsing, A. J., & McCauley, D. J. (2017). Extinction risk is most acute for the world's largest and smallest vertebrates. Proceedings of the National Academy of Sciences, 114, 10678– 10683.

Lines 173-176 – These results should be presented in the main manuscript.

Thanks for this comment. In order to make the manuscript more readable, for the covariates we decided to report the test values in the main text between brackets, and include tables and figures in supplementary. However, we are open to move them in the main text if the editors would advise this way.

Lines 182-186 – This isn’t fully described in the methods and appears to test the nominal CPUE with the same variables as in the other models but isn’t presented in a way that shows its added value to the results or discussed in detail in the manuscript.

Thanks for the comment, and we apologize for the lack of clarity in the previous version of the MS. The pRDA we run is a multivariate analysis including 6 response variables (the IUCN categories: CR, EN, VU, NT, LC, DD). The added value of the analysis is a result for each IUCN category, and this complement the univariate analyses carried-out on total NCPUE and BCPUE; we added a sentence to better explain that. Please see lines 200-204 (result and discussion in the clean file) and line 451-461 (methods in the clean file).

Line 215 – Just states CPUE, is this as you are referring to both NCPUE and BCPUE?

Thanks for this comment. We apologize for the lack of clarity. In the revised version we specified “BCPUE and NCPUE”.

Line 340-343 – Was it not possible to sex individuals from the photos to do species and sex specific size at maturity?

Thanks for the comment. Sex determination from photos was possible only in some instances, and it would not have been possible to have a consistent and reliable assessment for all the individuals. For this reason, we choose the conservative approach described in our study. We added a sentence in methods section to detail this point (Lines 368-369 in the clean file).

Lines 363-374 – From the description it looks like nominal NCPUE and BCPUE are being modelled as positive continuous variables and as such are these able to be considered as zero-inflated? Or are these approaches based on the zeros creating a skewed data set? This is also the analysis where an offset would best be used as this allows for the raw numbers of sharks being caught to be modelled as a rate per net length, and if zero-inflated, then ZIP Or ZINB models may be appropriate.

Thanks for this comment. As described in a previous answer to Reviewer #2, we used family distribution “Zero Adjusted Gamma (ZAGA)” for this specific analysis. ZAGA distribution consists of a two-part model: a binomial (*logit*, presence/absence) model (which predicts the probability of fishing operations with at least one elasmobranch individual in the catch) and a positive (truncated) abundance model (*log*, where all zeroes are excluded) (Stasinopoulos D. M. & Rigby R.A. 2007). The model with the ZAGA distribution (run on data standardized per unit of effort) was selected after comparing it with a ZIP on catch data not standardized per unit of effort, but including the offset (i.e. net length). AIC and the graphic inspection of the residuals were used to select the best approach and keep that in the MS. ZAGA was the best fit and we therefore selected it for this analysis. All these elements are provided in the main text, please see lines 394-413 (clean file). And as reported above, we, now, added a flow diagram in the Supplementary materials (Fig. 6S) to present the systematic approach we used to select the best model.

Lines 380-381 – What was the structure of the model for immature vs mature individuals? It is stated as using a ZAGA with an offset for net length. Was the offset the log of the net length? Are all the nets the same height? If not, perhaps m² is a better metric for effort? Also is a Gamma distribution the most suitable error structure for count data? The supplementary table for this analysis says the proportions of immature/mature individuals was modelled?

Thanks for the comment. We rephrased the sentence as it was confusing. We did not model immature vs mature but we modelled immature in PPAs vs UPAs and mature in PPAs vs UPAs for each IUCN categories (TH, NTH, DD); please see lines 414-419 (clean file). Nets were predominantly trammel nets (in about 95% of the fishing operations monitored) with a minority of gillnets (4%) and combined trammel-gill nets (1%) (lines 334-340 in the clean file). Trammel nets were the same height and the others nets were equally distributed between PPAs and UPAs. For this reason, we believe net-length is an informative measure, and can easily allow comparison with other study as this is largely used to estimate CPUE.

As specified above we followed a systematic approach for selecting the best model for each analysis. Please, find the flow diagram showing the selection of the best model in the supplementary materials (Fig. 6S). In particular, for the analyses about the mature and immature individuals, for each combination of IUCN category and state of maturity, we performed the GAMLSS analysis using both the error family distribution “Zero inflated Poisson” (ZIP) on count of immature and mature individuals (with the net length used as an offset) and the family distribution “ZAGA” on catch standardized per unit of effort, comparing the two each time to identify the best-fitting model. For each analysis we used the AIC results and the graphic inspection of the residuals to select the best approach and keep that in the MS.

Line 386 – Was there a formal test for zero-inflation? Plotting numbers/proportions of zeros doesn't inform us whether these are more than would be expected.

Thanks for spotting this, and we apologize for the lack of details in the previous version of the ms. We used zero-inflation model after a comparison between Poisson Vs. Zero-inflated Poisson models using the “Likelihood-ratio test”. We have now detailed this in the revised version of the ms (please see lines 394-407 in the clean file).

Lines 414-416 – What is the definition of a species composition response that is being tested in the pRDA?

Thanks for the comment, we apologize for the lack of clarity in the previous version of the ms. As the response variables were the IUCN categories, we changed the word “species” with “IUCN categories”.

Figures – I think figures for the outputs from the analysis of what is driving CPUE should be in the main manuscript.

Thanks for the comment. The study did not focus on the covariates but we used those only to disentangle the effect of protection, for this reason we reported test values in the main text and kept figures in supplementary in order to provide synthetic information in the manuscript and allow the readers to have additional details in supplementary. However, we are open to move them in the main text if the editor would advise this way.

Table S9 – There isn't an explanation that two model types were compared for this in the main text and so isn't clear what this table is showing. The description of the structure of the ZIP or ZAGA is not clear. If using proportion of immature versus mature, I'm not sure either of these error structures are suitable. The details of the model would be good here, similar to Table S10 to show what was being tested and structure of formula.

Thanks for this comment. The comparison between the two distribution (ZAGA and ZIP) is detailed now at lines 414-419 (clean file) and we added the flow-chart showing the selection of the model in the supplementary materials (Fig. 6S). In addition, we, clarified that immature and mature of each IUCN group (TH, NTH, DD) were separately analysed (6 different analyses per each distribution) to compare PPAs and UPAs. The variable tested in each model was only one (Protection: PPAs Vs. UPAs). Finally, we have also changed the table as requested.

GROUP	FAMILY ERROR	MODEL	AIC
NONTHREATENED (NTH)			
Immature	ZIP	modimm_zip<-gamlss(num ~ in.out+offset(log(net), family=ZIP)	448.19
Immature	ZAGA	modimm_zag<-gamlss(den ~ in.out, family=ZAGA)	445.50
Mature	ZIP	modmat_zip<-gamlss(num ~ in.out+offset(log(net), family=ZIP)	411.50
Mature	ZAGA	modmat_zag<-gamlss(den ~ in.out, family=ZAGA)	420.99
THREATENED (TH)			
Immature	ZIP	modimm_zip<-gamlss(num ~ in.out+offset(log(net), family=ZIP)	372.57
Immature	ZAGA	modimm_zag<-gamlss(den ~ in.out, family=ZAGA)	391.61
Mature	ZIP	modmat_zip<-gamlss(num ~ in.out+offset(log(net), family=ZIP)	117.28
Mature	ZAGA	modmat_zag<-gamlss(den ~ in.out, family=ZAGA)	117.54
DATA DEFICIENT (DD)			
Immature	ZIP	modimm_zip<-gamlss(num ~ in.out+offset(log(net), family=ZIP)	127.44
Immature	ZAGA	modimm_zag<-gamlss(den ~ in.out, family=ZAGA)	125.22
Mature	ZIP	modmat_zip<-gamlss(num ~ in.out+offset(log(net), family=ZIP)	98.01
Mature	ZAGA	modmat_zag<-gamlss(den ~ in.out, family=ZAGA)	98.47

Table S10 - It would read better if the models were presented from lowest AIC to largest, as these are how they are ranked. Why are some models in bold, it is not explained in the caption?

Thank you for this comment. Table S10 is structured following the output of StepGaic function that presents the models ranking from the largest to the lowest AIC, starting from the full model including all variables through the AIC selection (the most parsimonious model). We prefer to keep it as this is a standard way to present these results.

Some models were in bold due to typing mistake, we fixed it.

REVIEWERS' COMMENTS

Reviewer #2 (Remarks to the Author):

Thank you for your explanations and for adding that clarity into the text. For me, the text and analyses are a lot clearer and it allows for the importance of your work to be understood with ease. I have very minor suggestions below with regards to text editing.

Line 102: 737,71 km – missing a value or comma is misplaced

Line 109 Threatened [TH} – IUCN usually uses THR for threatened.

Line 113 – should common guitarfish and others be capitalized? Some species are capitlized and some are not.

Line 122: In terms of Biomass, TH species were the most caught. I suggest revising this to: The greatest biomass per unit effort was of TH species.

Line 152: change to non-threatened

Line 203: capitalized the IUCN categories and remove ``

Reviewer #3 (Remarks to the Author):

Thank you for your responses and clarity throughout the manuscript, I look forward to seeing it in press.

Reviewer #2:

Thank you for your explanations and for adding that clarity into the text. For me, the text and analyses are a lot clearer and it allows for the importance of your work to be understood with ease. I have very minor suggestions below with regards to text editing.

Thank you very much for appreciating our work

Line 102: 737,71 km – missing a value or comma is misplaced

Done

Line 109 Threatened [TH] – IUCN usually uses THR for threatened.

Thank you for the suggestion. We modify the term accordingly

Line 113 – should common guitarfish and others be capitalized? Some species are capitlitzed and some are not.

Thank you for spotting this, the species' names were capitalized

Line 122: In terms of Biomass, TH species were the most caught. I suggest revising this to: The greatest biomass per unit effort was of TH species.

Thank you for the suggestion, we modify the sentence accordingly

Line 152: change to non-threatened

Thank you. Done

Line 203: capitalized the IUCN categories and remove ‘ ‘

Done. Thank you.

Reviewer #3:

Thank you for your responses and clarity throughout the manuscript, I look forward to seeing it in press.

We thank the reviewer for appreciating our work.